# Added value of seasonal hindcasts to create UK hydrological drought storylines

Wilson C.H. Chan[1,3], Nigel W. Arnell[1], Geoff Darch[2], Katie Facer-Childs[3], Theodore G. Shepherd[1], Maliko Tanguy[3]

[1]Department of Meteorology, University of Reading, Reading, UK
[2]Anglian Water, Peterborough, UK
[3]UK Centre for Ecology & Hydrology (UKCEH), Wallingford, UK

*Correspondence to*: Wilson Chan (wilcha@ceh.ac.uk)

**Abstract.**

The UK has experienced recurring periods of hydrological droughts in the past, including the drought declared in summer 2022. Seasonal hindcasts, consisting of a large sample of plausible weather sequences, can be used to create drought storylines and add value to existing approaches to water resources planning. In this study, the drivers of winter rainfall in the Anglian region in England are investigated using the ECMWF SEAS5 hindcast dataset, which includes 2850 plausible winters across 25 ensemble members and 3 lead times. Four winter clusters are defined using the hindcast winters based on possible

combinations of various atmospheric circulation indices (such as North Atlantic Oscillation (NAO), East Atlantic (EA) pattern and the El-Niño Southern Oscillation). Using the 2022 drought as a case study, we demonstrate how storylines representing alternative ways the event could have unfolded can be used, to explore plausible worst cases over winter 2022/23 and beyond. The winter clusters span a range of temperature and rainfall response in the study region and represent circulation storylines that could have happened over winter 2022/23. River flow and groundwater level simulations with the large sample of plausible

hindcast winters show that drier than average winters characterised by predominantly NAO-/EA- and NAO+/EA- circulation patterns could have resulted in the continuation of the drought with a high likelihood of below normal to low river flows across all selected catchments and boreholes by spring and summer 2023. Catchments in Norfolk were particularly vulnerable to a dry summer in 2023 as river flows were not estimated to recover to normal levels even with wet winters characterised predominantly by NAO-/EA+ and NAO+/EA+ circulation patterns, due to insufficient rainfall to overcome previous dry

conditions and the slow response nature of groundwater-dominated catchments. Through this analysis, we aim to demonstrate the added value of this approach to create drought storylines during an ongoing event. Storylines constructed in this way supplement traditional weather forecasts and hydrological outlooks to explore a wider range of plausible outcomes.

## 1 Introduction

The United Kingdom has experienced recurring periods of hydrological droughts in the past (Marsh et al. 2007; Barker et al.

2019). This includes the latest drought declared in summer 2022 (Parry 2022). Planning against a wide range of plausible

outcomes is of interest to water resources planners. Water companies are required to outline demand and supply management actions in their drought plans and to prepare for the application of drought permits and drought orders during an event (Defra 2021). Water companies can enact several measures during a drought, including demand management measures such as temporary use bans (TUBs) and non-essential use bans (NEUBs). Should a drought worsen and river flows remain low, water

companies must demonstrate serious deficiency of supplies due to exceptional shortage of rainfall to apply for drought permits and drought orders to enable continued abstraction as detailed in water company drought plans (e.g. Anglian Water Drought Plan 2022).

The approach taken to plan for ongoing events differs between water companies but existing approaches can be separated into

40 three strands. First, weather forecasts are used for operational drought forecasting. For example, the ECMWF SEAS5 forecasts provide seasonal (up to 215 days) forecasts which can be used as input to hydrological models at a small number of catchments. Previous studies have highlighted that probabilistic forecasts have limited skill in the forecasting of meteorological droughts on a sub-seasonal and seasonal timescale as precipitation is challenging to predict at long lead times (e.g. Richardson et al. 2020). Other forecasting methods that do not use probabilistic rainfall forecasts are currently operationally applied. These

45 include persistence forecasts based on flow anomaly in the most recent month and historical analogue forecasts using the most similar historical sequences as employed in the UK Hydrological Outlook (Svensson 2016). Second, trajectories of river flows can be created through catchment hydrological model simulations assuming that rainfall over the next month reaches a specific percentage of long term average (LTA) rainfall (e.g. 60%, 80% or 100% of LTA). This approach is taken by the Environment Agency in their monthly water situation reports (e.g. Environment Agency 2022). Third, forecasts can be provided using

information from historical climate, either by assuming the repetition of individual notable historical years (such as benchmark events like 1975-76 or La Niña drought years like 2011) or by repeating all available years in an Ensemble Streamflow Prediction (ESP) approach such as that taken by the UK Hydrological Outlook (Prudhomme et al. 2017; Harrigan et al. 2018). The second and third strands are representative of a storyline approach which aims to describe and quantify pathways and developments of past or future events conditioned on changes to the event's drivers (Shepherd et al. 2018). However, both

strands are subject to some drawbacks. The relatively simple assumption of rainfall as a percentage of long term average does not consider physical plausibility and cannot be traced to climate drivers, while the ESP approach is hampered by the limited length of observations, constraining the range of outcomes.

River flows, groundwater aquifers and reservoirs in the UK are often replenished from winter rainfall. Winter rainfall is

60 determined by atmospheric circulation, usually the variability in position and strength of the North Atlantic jet stream. Previous research has found a positive correlation between the North Atlantic Oscillation (NAO) and winter rainfall for western UK with more storms over northern Europe when the NAO is in its positive phase (e.g. Svensson et al. 2015; Hall and Hanna 2018). For southeast and eastern England, studies have found that variability of winter rainfall arises from the combined influence of various circulation indices, particularly the East Atlantic (EA) pattern which can either enhance or dampen the

surface temperature and rainfall response to the NAO (Mellado-Cano et al. 2019; West et al. 2021). Previous research by Emerton et al. (2017) has highlighted the role of El Niño Southern Oscillation (ENSO) in streamflow variability showing that low flows during winter months across the UK are more likely during La Niña. This is because ENSO can influence the NAO and other circulation anomalies which can lead to negative winter rainfall anomalies in southern England, with past multi-year meteorological droughts featuring dry winters often coinciding with La Niña conditions (Folland et al. 2015). In particular, East Anglia is particularly vulnerable to prolonged dry weather and droughts as it is the driest region in the UK, receiving only 71% of the average UK annual rainfall (Anglian Water Drought Plan 2022). The specific aims of this study are to:

- Investigate the drivers of winter rainfall for the region of eastern England supplied by Anglian Water using a large sample of plausible winters from the ECMWF seasonal forecasting system SEAS5;
- Use knowledge of winter rainfall drivers (e.g. atmospheric circulation patterns) to group SEAS5 hindcast winters into conditional storylines representing plausible winter rainfall trajectories based on various combinations of atmospheric circulation patterns;
- Create drought storylines of river flows and groundwater levels for the 2022 drought representative of what the drought could have looked like if winter 2022/23 resembled the hindcast winters within each winter circulation storyline. The mean deficit and maximum intensity of each drought storyline are calculated using simulated river flows.

Note that this study is not attempting to verify or assess the skill of SEAS5 in forecasting rainfall and droughts, which has been the focus of previous studies (e.g. Haiden et al. 2018; Arnal et al. 2018; Johnson et al. 2019). Indeed, this study is not about forecasting at all. Instead, the aim is to use the seasonal hindcasts across past winters — which are constructed for the purpose of calibrating seasonal forecasts — to explore the utility of the storyline approach as a complementary tool to traditional forecasting methods, to assist water resources planning. Statistical tests on range, stability and independence are conducted in Section 2 to confirm the reliability of the SEAS5 hindcasts for the aims of this study.

## 2 Data and methods

### 2.1 Hindcast data and model fidelity

The SEAS5 hindcast dataset (1982-2021) is used to provide a large sample of plausible winters (Dec, Jan, Feb - DJF). In total, there are 2850 winters in the hindcast dataset across 25 ensemble members and three lead times (Sep, Oct and Nov) (comprising 38 complete winters between 1983 and 2020 x 25 ensemble members x 3 lead times). SEAS5 ensemble members are generated by perturbations made to their initial atmospheric and oceanic conditions (Johnson et al. 2019). The mean of the modelled rainfall is scaled to match observed mean winter rainfall at the catchment scale for each selected catchment. In this study, the large sample of hindcast winters are treated as individual plausible outcomes in an approach that follows the UNSEEN

technique in Thompson et al. (2017) where a large ensemble of model simulations is used to explore plausible events that could have happened as well as unprecedented events. The pooling of initialised ensembles is widely employed to expand the sample size available in order to explore a wider range of possible outcomes (e.g. van den Brink et al. 2004; Kelder et al. 2021;

Brunner and Slater 2022, among others). Figure 1 shows observed mean winter rainfall for the Greater Anglia region over the 1983-2016 period from the CEH-GEAR dataset and the spread of scaled modelled rainfall from the hindcasts across all ensemble members and lead times over the same period. The observational data points fall within the interquartile range of modelled rainfall for roughly one-half of the years and fall within the maximum and minimum in all years. Standardising the observed rainfall against the modelled rainfall distribution further shows that for 22 out of the 38 years, the observed value is

within one standard deviation of the modelled rainfall of that year. The SEAS5 hindcasts also do not seem to be exaggerating the likelihood of low rainfall as 4 out of the 38 observed winters fall within the lowest 10% of the standardised modelled rainfall distribution. There is a 75% chance of this occurring according to the binomial probability formula. Together this shows that the SEAS5 hindcasts are reliable and useful for the representation of the rainfall climatology for the study region over the selected period.

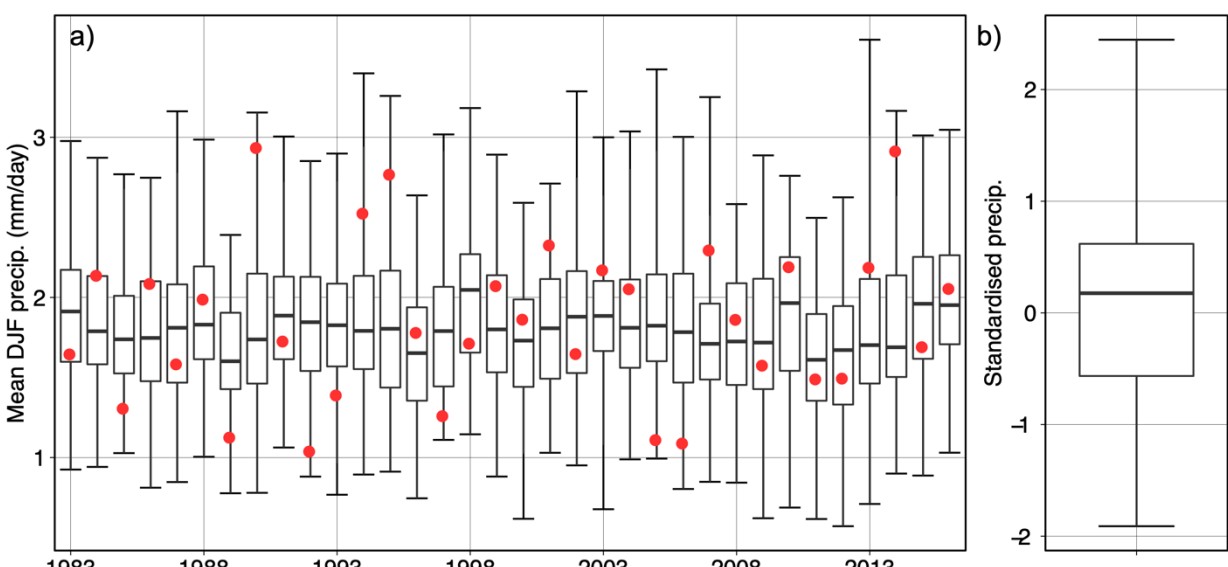

**Figure 1 (a) Observed (CEH-GEAR) and simulated mean winter (DJF) rainfall over 1983-2016 from the SEAS5 hindcasts across 25 ensemble members and three lead times over the East Anglia region. The whiskers of the boxplots extend to the maximum and minimum modelled value. (b) Distribution of observed winter rainfall for each year over the 1983-2016 period standardised against**

**the distribution of simulated rainfall of the same year.**

The credibility of the hindcast winters is further tested using the model fidelity test in Thompson et al. (2017) by comparing the statistical moments of observed rainfall with 10,000 subsamples of the SEAS5 modelled rainfall (Figure 2a for an example

catchment). SEAS5 winters over the East Anglia region are deemed statistically indistinguishable from the observations as the

120 four statistical moments of the observed winter rainfall lie within 95% of the distribution of the respective statistical moments from the subsamples of hindcast winters. Whilst this is guaranteed for the mean due to the scaling, it is not necessarily the case for the three other moments. Additional tests on ensemble member stability and independence were conducted following Kelder et al. (2020, 2022) Stability refers to the potential for ensemble members to drift towards their (biased) climatology from their observation-based initial conditions, and can be assessed by comparing the distribution of simulated variables across

lead times. As seen from Figure 2b, the distribution of simulated winter rainfall is similar across the three lead times, indicating that there is no discernible drift. Independence refers to whether individual ensemble members for each lead time are independent of each other, which can be assessed by calculating the Spearman rank correlation of modelled rainfall for every distinct pair of ensemble members. As seen from Figure 2c, the median correlation for all three lead times is close to zero, showing that the ensemble members can be considered independent from each other over the considered timescale.

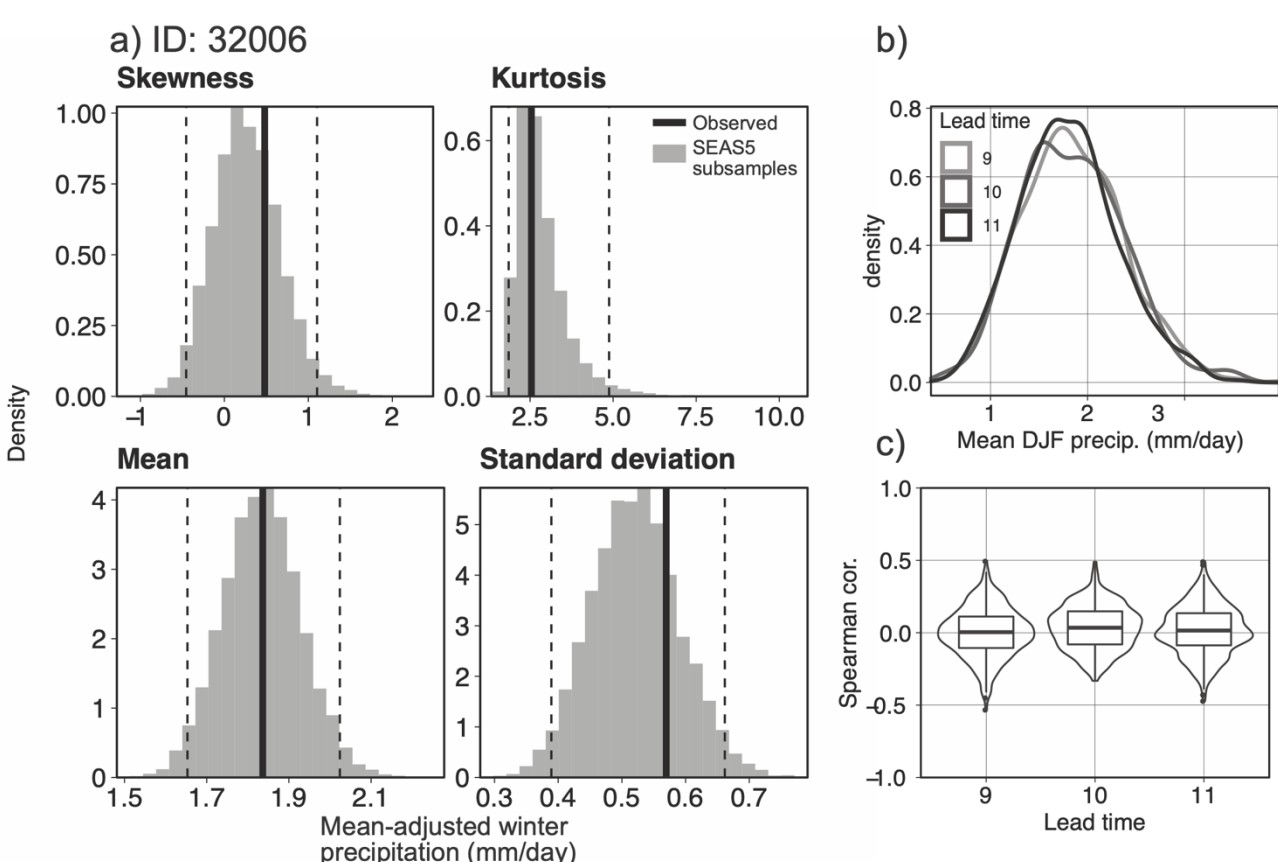

**Figure 2: Example of the model fidelity test for one catchment (ID: 32006). a) The statistical moments of observed DJF rainfall and of 10,000 subsamples of the modelled rainfall in SEAS5. Hindcast winters are scaled to match the mean observed rainfall. b) Test for model stability and c) ensemble member independence across three lead times.**

## 2.2 Drivers of dry winters

A series of meteorological indices describing atmospheric circulation patterns are calculated using monthly mean sea level pressure anomalies in the European/North Atlantic region from both observed winters (ERA5 reanalysis - 1960-2015) and for each winter in the hindcast dataset. The NAO index and the EA indices are represented by the first two leading modes calculated through empirical orthogonal function (EOF) analysis. The Niño3.4 index is calculated from average sea surface temperature anomalies in the region (5°S-5°N, 120-170°W) to represent the phase of ENSO. Figures 3(a)-(c) show the relationship between rainfall anomalies over East Anglia for both observed and hindcast winters with the average winter Niño3.4, NAO and EA index. There is no clear relationship between ENSO phase and rainfall anomalies. There is a weak negative relationship between the NAO index and rainfall anomalies and a positive relationship between the EA index and rainfall anomalies. For each winter between 1982 and 2020, there are 75 simulated winters in the hindcasts across ensemble members and lead times. There is considerable variability in the NAO and EA phases across the hindcast winters each year which often spans the four possible combinations of NAO and EA (hence the high variability in rainfall anomalies). Conversely, there is little variability in ENSO phase across the hindcast winters for each year as ENSO is comparatively slowly evolving and hence more predictable several months ahead. For example, winters 2015/16, 1997/98 and 1982/83 all exhibited particularly strong El Niño conditions and the hindcasts issued prior to each of those winters all had a similarly high Niño3.4 index value (as shown by the vertical cluster of points in Figure 3a). Figure 3d shows the rainfall anomalies associated with combinations of NAO and EA phases. NAO+/EA- are most likely associated with drier than average conditions and NAO-/EA+ winters with wetter than average conditions although there are also outliers with notable dry winters in both the observations and the hindcast. The relationships between rainfall and meteorological indices for the hindcast winters are consistent with past work showing the influence of opposing phases of the NAO and EA on observed UK rainfall (e.g. West et al. 2021, 2022; Parry et al. 2012).

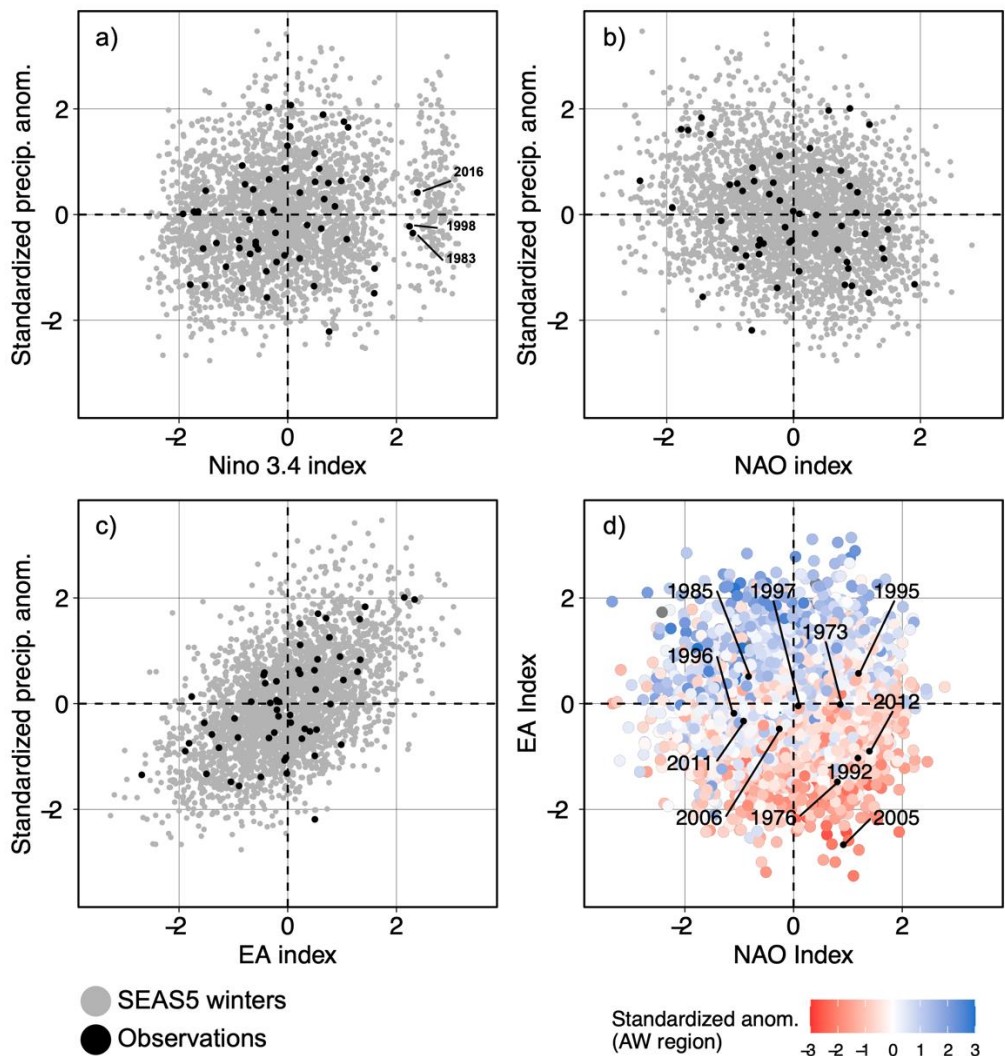

**Figure 3:** Relationship of a) Niño3.4, b) NAO, and c) EA index with standardised rainfall anomalies in East Anglia for the SEAS5 winters (grey) and observed winters (black). Panel d) shows the NAO index and EA index for each SEAS5 winter together with the rainfall anomalies over East Anglia associated with each winter (colours). The cluster of strong El Niño hindcast winters in panel a) relate to hindcast winters issued prior to winters 2015/16, 1997/98 and 1982/83 (black dots in panel a), which were three of the strongest El Niño winters in the observations. Selected dry winters in the observations are shown in panel d) by the black dots and the year labels.

## 2.3 Meteorological indices and circulation storylines

In addition to the NAO, EA and ENSO, additional indices are calculated to explore polar vortex strength (defined as average wind speed (U10) at 10hPa and 60°N) and sea surface temperatures (SST tripole index – Fan and Schneider 2012). K-means

clustering of all the calculated indices is used to create clusters with similar characteristics. Winter clusters are created separately for La Niña and El Niño winters but subsequent hydrological modelling was only completed for La Niña winters given the La Niña conditions observed over 2022. Figure 4 shows four clusters defined for La Niña winters in the hindcast. The clusters show little difference between El Niño and La Niña for clusters 1 and 3 but La Niña winters are in general drier than El Niño winters for clusters 2 and 4 (Figure S1), which are associated with NAO+ conditions and a strong polar vortex (Figure 3). Temperature anomalies associated with each cluster are shown in Figure S2. Four clusters are chosen as they primarily reflect the four possible combinations of opposing phases of the NAO and EA which have been shown to have distinct signatures for rainfall over southeast England (West et al. 2021), including where opposing phases of the EA pattern may reverse the rainfall signal given a particular NAO phase (Mellado-Cano et al. 2019). The clusters also consider the range of circulation response and climate anomalies such as changes in polar vortex strength. Using the same SEAS5 hindcasts, Kolstad et al. (2022) showed the wide range of winter surface temperature responses that can arise from a given vortex state due to confounding factors such as NAO and ENSO. Clusters 1 and 2 (3 and 4) are generally associated with drier (wetter) than average rainfall over East Anglia although drier than average winters can occur for all clusters. This is exemplified by the fact that notable dry winters in the observations have exhibited characteristics of all four clusters (Figure 5). The observed dry winters of 2011/12 and 1975/76 closely resemble the composite mean sea level pressure anomalies of clusters 1 and 2, respectively. For clusters 3 and 4, the composite mean shows low pressure over the British Isles, resulting in generally wetter winters. Observed winter 1984/85 resembles cluster 3 but with an extension of the high pressure eastwards with the pressure centre over Scandinavia leading to drier than average conditions in eastern England. Similarly, winter 1972/73 resembles the composite mean for cluster 4 but with a northward extension of the high pressure over southern UK.

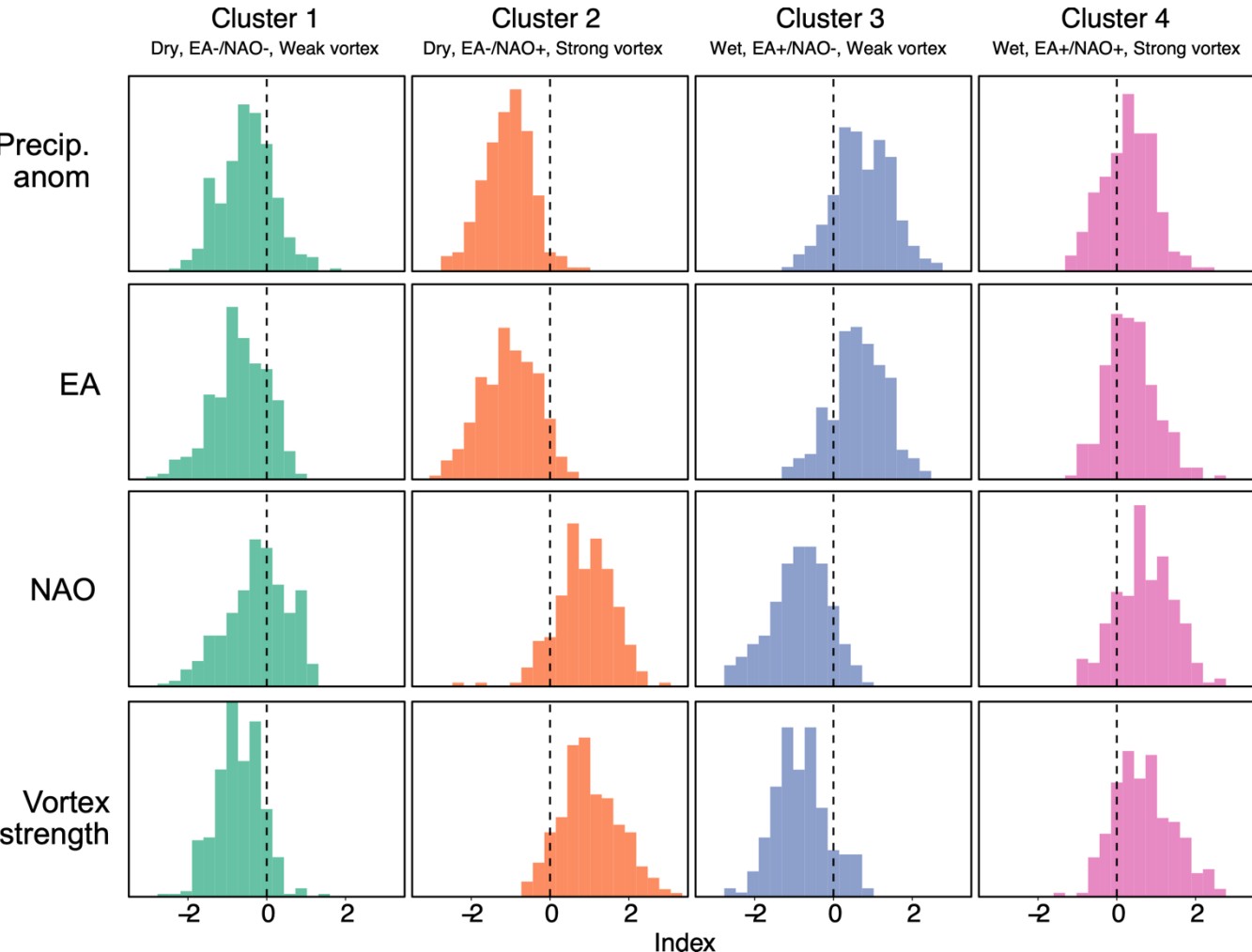

**Figure 4: Winter clusters defined from hindcast winters with La Niña conditions using k-means clustering and the standardised rainfall anomalies and meteorological indices associated with each cluster.**

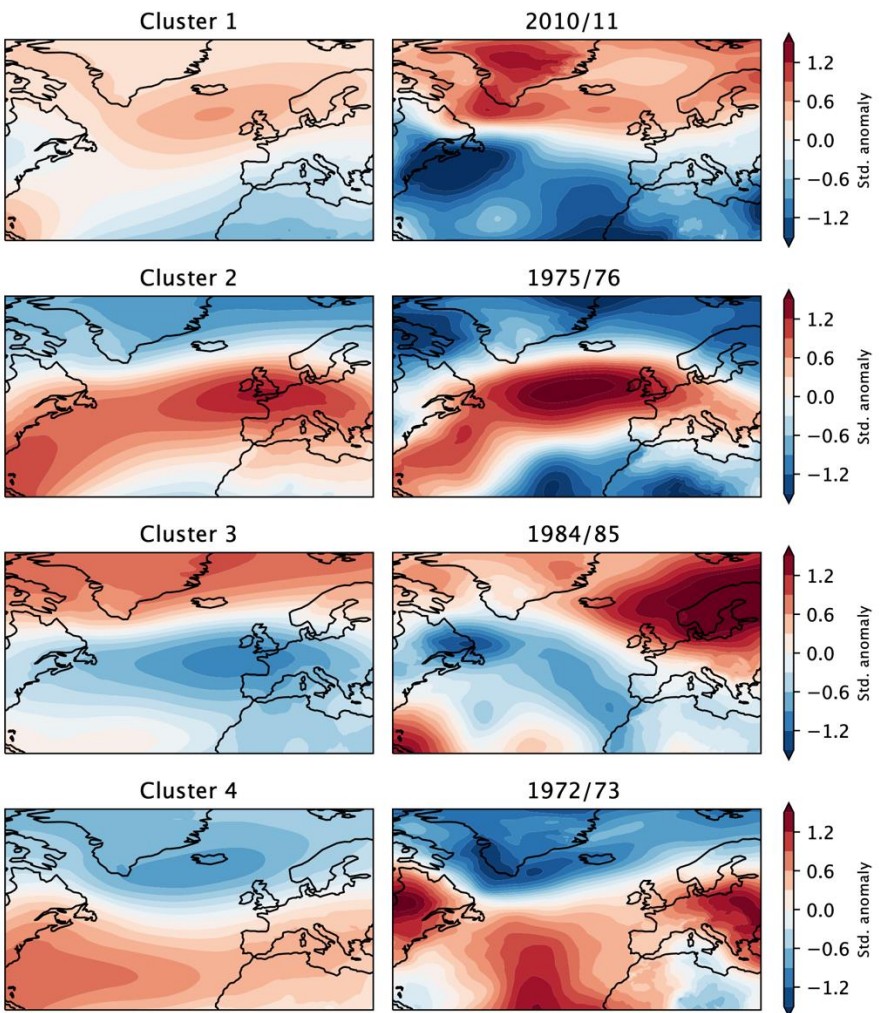

**Figure 5: Composite mean sea level pressure (SLP) anomalies (relative to ERA5 1960-2015) for the four circulation storylines and SLP anomalies for selected dry winters from ERA5 associated with similar patterns for each cluster.**

## 2.4 Study catchments and hydrological modelling

The GR6J hydrological model was used to simulate river flows at 16 river catchments within East Anglia (Table 1). The selected catchments include key abstraction catchments and catchments that supply reservoirs operated by Anglian Water. GR6J is a lumped catchment hydrological model with six parameters for calibration and is designed to improve low flow 200 simulation, particularly over groundwater-dominated catchments (Pushpalatha et al. 2011). Aquimod, a lumped groundwater level model developed by the British Geological Survey (Mackay et al. 2014), was used to simulate groundwater levels at 10

boreholes (Table 2). Both GR6J and Aquimod are increasingly used in research and industry, including by Anglian Water operationally for their drought forecasts and long-term water resources planning (Anglian Water Drought Plan 2022).

Daily observed mean catchment-averaged rainfall (CEH-GEAR dataset – Tanguy et al. 2021) and potential evapotranspiration (PET) calculated from temperature (HadUK-Grid dataset – Hollis et al. 2019) were used to drive the hydrological model over the baseline period (1982-2014) for model calibration. PET was calculated following the method presented in Tanguy et al. (2018) using the McGuinness-Bordne temperature-based equation calibrated for the UK. A temperature-based method to calculate PET was chosen as such methods are relatively simple to apply and are regularly used by water companies for drought

forecasting. The standardised streamflow index (SSI) was calculated for each storyline to consider drought intensity. SSI over various accumulation periods was calculated by fitting a Tweedie distribution to monthly simulated river flows following Svensson et al. (2017). GR6J was calibrated individually for each catchment across the baseline period via the multi-objective approach set out in Smith et al. (2019). In short, 10,000 parameter sets were generated via Latin Hypercube Sampling and ranked based on model performance metrics that compare simulated and observed river flows. The model performance metrics

selected include Nash-Sutcliffe Efficiency (NSE), NSE of logarithmic flows, absolute percentage bias, percentage error in Q95 (low flows) and percentage error in mean annual minimum (30-day averaged) flow. Model performance statistics for all metrics are provided in the supplementary materials (Figure S3). Observed and simulated river flows at six example catchments over the baseline period and observed and simulated standardised streamflow index accumulated over 3 months (SSI-3) are shown in Figures S4 and S5 respectively.  Aquimod was driven by rainfall and PET averaged across the closest 40km MORECS grid

to the borehole location (Hough and Jones 1997) in the same way as employed operationally by Anglian Water. A Monte Carlo parameter sampling approach was used to generate a random set of parameters and model performance was assessed using NSE (Mackay et al. 2014). The model was previously calibrated for the selected boreholes (Bunting et al. 2020) and the top parameter set from that study was used here.

**Table 1: Details of the selected catchments within the study region. The National River Flow Archive (NRFA) station id, station name, latitude, longitude, baseflow index (BFI) and the Nash-Sutcliffe Efficiency of logarithmic flows (logNSE) for the top performing parameter set are provided.**

| NRFA station id | Station | Latitude | Longitude | Baseflow Index | logNSE |
|---|---|---|---|---|---|
| 37024 | Colne at Earles Colne | 51.94 | 0.70 | 0.43 | 0.69 |
| 31007 | Welland at Barrowden | 52.59 | -0.60 | 0.50 | 0.76 |
| 33026 | Bedford Ouse at Offord | 52.29 | -0.22 | 0.51 | 0.75 |
| 37005 | Colne at Lexden | 51.90 | 0.85 | 0.52 | 0.78 |
| 31010 | Chater at Fosters Bridge | 52.62 | -0.58 | 0.53 | 0.82 |

| | | | | | |
|---|---|---|---|---|---|
| 33035 | Ely Ouse at Denver Complex | 52.58 | 0.34 | 0.57 | 0.68 |
| 32006 | Nene at Upton Total | 52.23 | -0.95 | 0.58 | 0.82 |
| 32010 | Nene at Wansford | 52.58 | -0.41 | 0.60 | 0.84 |
| 34014 | Wensum at Swanton Morley Total | 52.73 | 0.99 | 0.75 | 0.88 |
| 34004 | Wensum at Costessey Mill | 52.67 | 1.22 | 0.76 | 0.81 |
| 33019 | Thet at Melford Bridge | 52.41 | 0.76 | 0.78 | 0.83 |
| 33006 | Wissey at Northwold Total | 52.54 | 0.61 | 0.82 | 0.84 |
| 34011 | Wensum at Fakenham | 52.83 | 0.85 | 0.82 | 0.84 |
| 33029 | Stringside at Whitebridge | 52.58 | 0.53 | 0.84 | 0.76 |
| 33007 | Nar at Marham | 52.68 | 0.55 | 0.90 | 0.82 |
| 29003 | Lud at Louth | 53.37 | 0.001 | 0.90 | 0.74 |

**Table 2: Details of the selected groundwater boreholes within the Anglian Water region. The Environment Agency code, borehole name, latitude, longitude and the Nash-Sutcliffe Efficiency (NSE) score for the top performing parameter set are provided. NSE is calculated over the period when observational records area available and based on Bunting et al. (2020).**

| Environment Agency code | Observation borehole | Latitude | Longitude | MORECS grid | NSE |
|---|---|---|---|---|---|
| TM04/695 | Castle Farm | 52.10 | 1.01 | 141 | 0.79 |
| TL65/050 | Dullingham | 52.21 | 0.36 | 140 | 0.78 |
| 1/610 | Grange de Lings | 53.29 | -0.53 | 108 | 0.76 |
| 5/108 | Horkstow Rd Barton | 53.67 | -0.45 | 101 | 0.59 |
| 2/544 | Leasingham | 53.02 | -0.43 | 118 | 0.71 |
| TG13/765A | Old Hall Thurgarten | 52.89 | 1.23 | 120 | 0.51 |
| TL66/094 | Springhead Farm | 52.25 | 0.34 | 140 | 0.70 |
| 2/566 | Stow to Oakholt | 53.21 | -0.43 | 109 | 0.80 |
| TL76/110 | Tank Hall | 52.26 | 0.54 | 140 | 0.77 |
| TF 81/010 | Washpit Farm | 52.74 | 0.69 | 130 | 0.78 |

In this study, storylines were created in autumn 2022 without prior knowledge of the winter 2022/23 (except that ENSO was known to be in a La Niña phase) to represent plausible pathways of the 2022 drought assuming winter 2022/23 resembled 235 winters within each of the four clusters. We aim to demonstrate the added value of this approach to explore a wide range of

plausible outcomes during an ongoing event. A brief exploration of the observed winter 2022/23 is provided in the discussion (Section 4). While there have been advances in probabilistic forecasts, plausible worst cases will by definition lie in the tail of the distribution and their likelihood will not be well represented by finite-sized ensembles. Understanding plausible worst cases by pooling hindcasts and treating each hindcast winter as individual plausible outcomes can be valuable as a "perfect"

probabilistic forecast may not be attainable. Storylines were simulated by running GR6J and Aquimod using the top parameter set for the baseline period up until November 2022 after which hindcast rainfall and PET data for each winter (DJF) in the four winter clusters were appended in place of winter 2022/23. Following the procedure employed by Anglian Water for operational drought forecasting, the hindcast winter rainfall was bias-adjusted for each catchment using quantile mapping (initiated via the *qmap* R package – Gudmudsson 2016). Spring (MAM), summer (JJA) and autumn (SON) 2023 were assumed to have 100%

long term average (LTA) rainfall by selecting the closest years matching 100% LTA rainfall in the observations. To understand the importance of winter rainfall and the effect of a second consecutive dry summer, an additional sensitivity test assumed summer (JJA) 2023 to follow 60% LTA seasonal rainfall.

## 3 Results

### 3.1 Storylines of the 2022 drought

#### 3.1.1 River flows

The 2022 drought is characterised by a dry spring-summer sequence (51% of LTA MAMJJA rainfall in East Anglia). The drought also followed an unusual pattern of rainfall in winter 2021/22 with average rainfall in December 2021, settled and dry conditions in January 2022 and wetter than average conditions in February 2022. Total winter rainfall was slightly below normal (97% of LTA) with drier conditions concentrated in the southeast of the region (e.g. southeast Suffolk). Exceptional

soil moisture deficits during the summer 2022 heatwave also exacerbated agricultural drought conditions. East Anglia experienced slightly above average rainfall in autumn 2022 (117% of LTA) which saw recovery of river flows at some catchments. Above average rainfall was mostly concentrated in western parts of the region with river flows in the northeast remaining below normal entering winter 2022/23 (Environment Agency 2022). Figure 6 shows simulated river flow response over winter 2022/23 for each circulation storyline. All catchments were estimated to experience below normal river

flows when entering spring 2023 given winters in clusters 1 and 2 with particularly severe flow deficits in groundwater-dominated catchments in the northeast of the region. Despite the wetter weather for winters in clusters 3 and 4, the groundwater-dominated catchments in the northeast were still estimated to experience below normal to low flows by spring 2023. This was likely due to the combined effect of insufficient winter rainfall to overcome dry conditions and the slow response nature of groundwater-dominated catchments. The outlook for each storyline is contrasted with the unclustered

outlook of flows across all 2850 winters (Figure S6). Using all 2850 winters highlights the confidence in the expectation of below normal flows in the northeast, but does not consider the dynamical drivers of winter rainfall that could lead to a

different likelihood of possible flow response as shown in the conditional subsets of each storyline. Figure 7 shows simulated river flows over winter 2022/23 from the standard ESP approach and for each circulation storyline for two example catchments. Compared to ESP, which assumes the repetition of historical years in limited observations, the use of a larger sample of hindcast winters explores a wider range of plausible outcomes. This includes outcomes that are outside the range obtained from the ESP approach, including both low (e.g. Cluster 2 outcomes for catchment 34014) and high flows (e.g. Cluster 3 outcomes for catchment 33019).

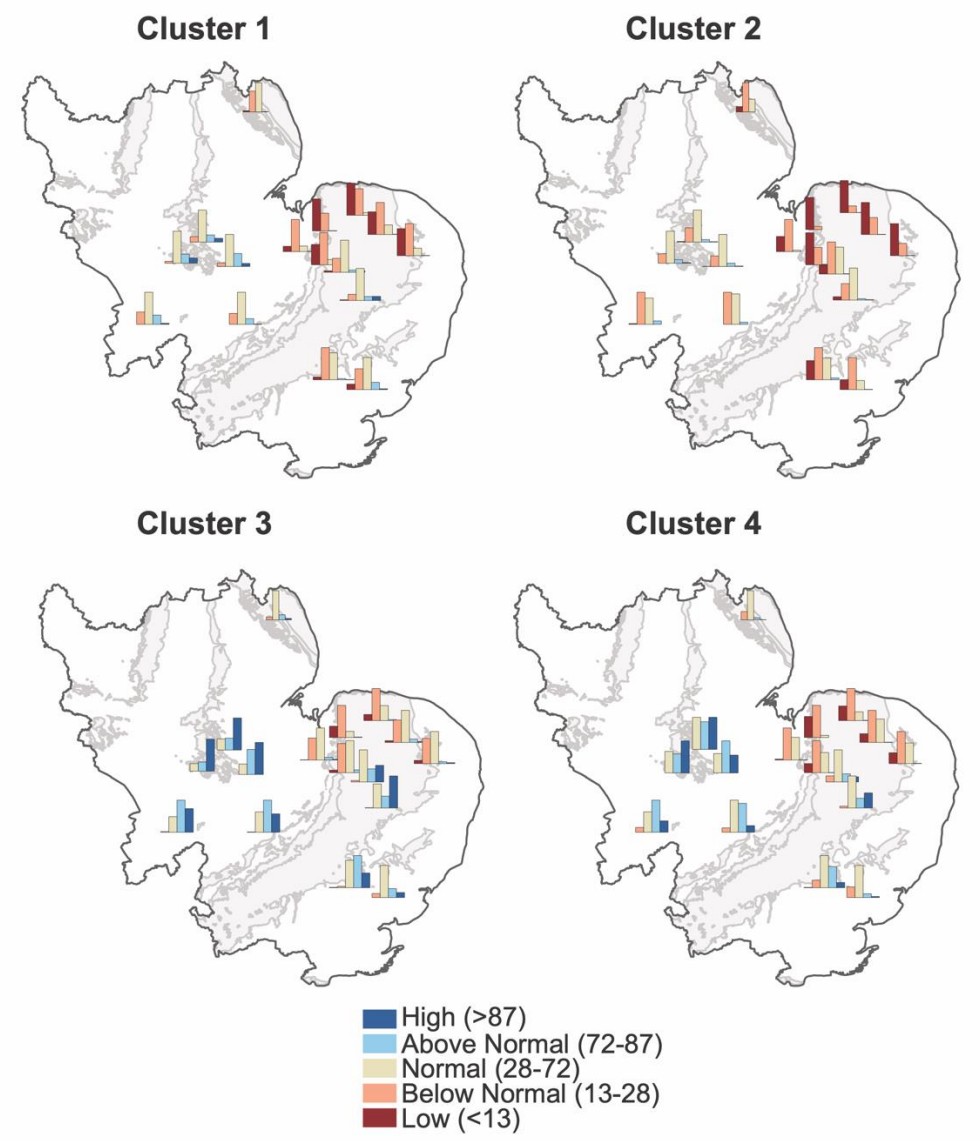

Figure 6: Outlook of river flows for each storyline represented in percentile terms relative to 1965-2015. Each storyline assumes winter 2022/23 follows hindcast winters in one of the La Niña winter clusters. Individual plots show the distribution of hindcast

winters for each percentile category as indicated by the colour key. Grey shading shows major aquifers in eastern England from the hydrogeology map of the British Geological Survey (https://www.bgs.ac.uk/datasets/hydrogeology-625k/).

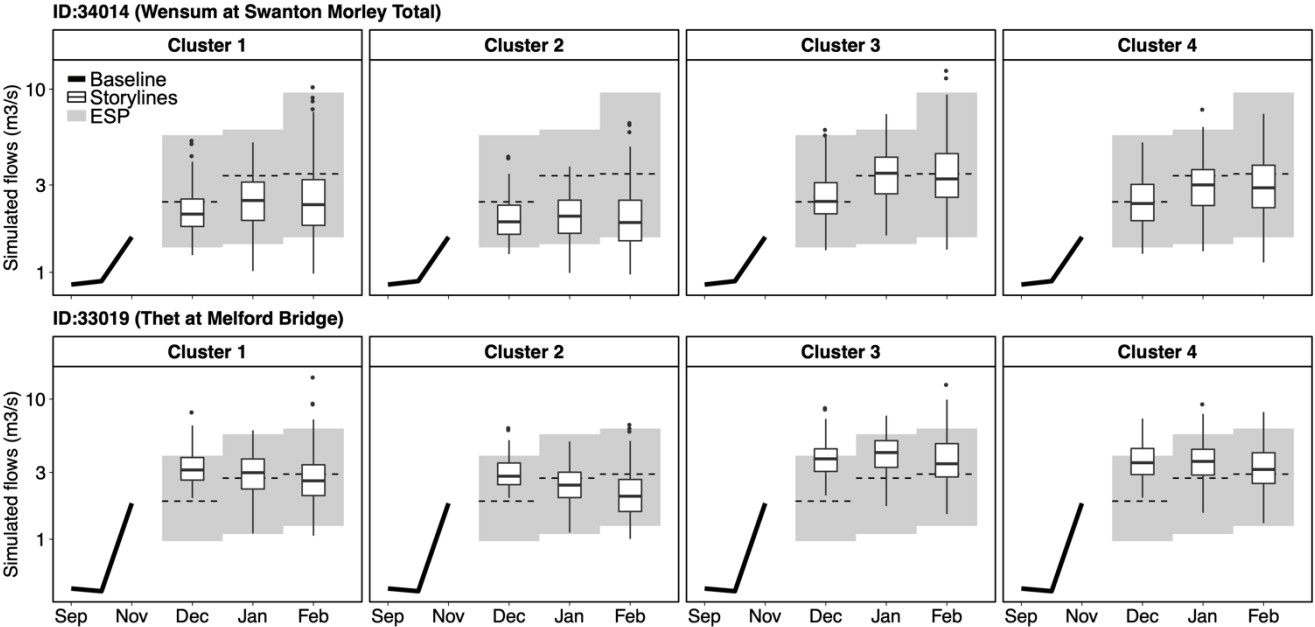

**Figure 7 Simulated river flows at two example catchments for each circulation storyline over winter 2022/23. The shading indicates the range of outcomes obtained from the standard ESP method by assuming the repetition of historical years (1965-2017) and the dotted line is the mean of the ESP outcomes.**

Figure 8 compares drought evolution (characterised by the standardised streamflow index accumulated over 3 months – SSI-3) during 1975-76 and 2021-22 with storyline estimates of SSI-3 for 2023 for four example catchments. Similar to 2022, 1976 was characterised by a dry spring-summer sequence (50% LTA rainfall in the study region). The decline to drought conditions in 2022 was generally later in the year and less severe with river flows generally recovering later in the autumn compared to 1975-76. Outlooks from the driest (cluster 2) and wettest (cluster 3) storylines show the continued vulnerability of catchments in East Anglia in 2023. For groundwater dominated catchments such as the Nar at Marham (33007) and Ely Ouse at Denver Complex (33035), drought intensity could plausibly match that seen in summer 1976 by summer 2023 given a dry winter in cluster 2 (mostly associated with NAO+/EA-). For these catchments, drought conditions could plausibly decline to similar drought intensity as seen in summer 2022 even with a wet winter in cluster 3 (mostly associated with NAO-/EA+).

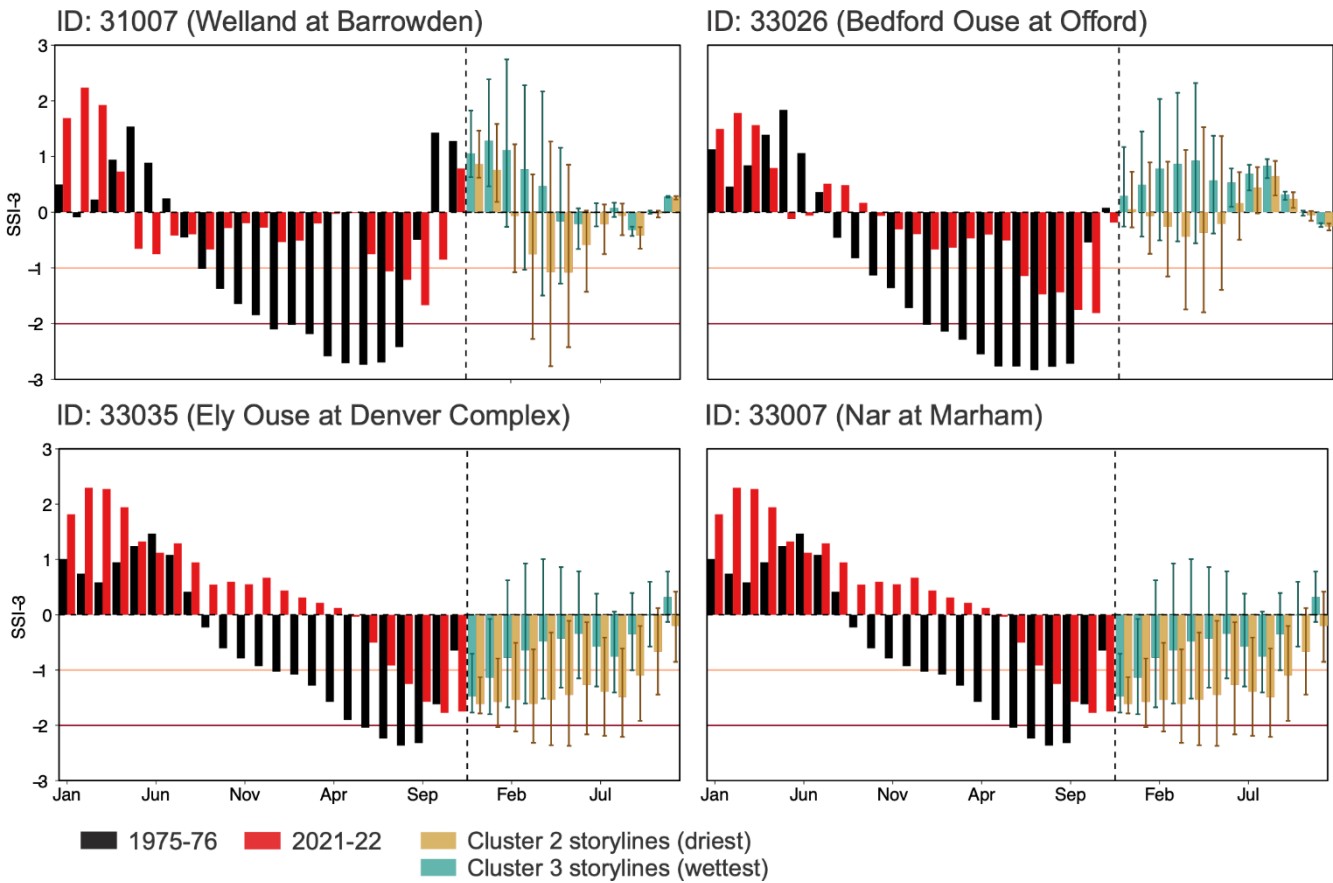

**Figure 8: Standardised streamflow index accumulated at 3-months (SSI-3) over 2021-2022 and beyond following the driest and wettest circulation storyline at four example catchments compared with SSI-3 over 1975-1976. Spring to autumn 2023 is assumed to have 100% LTA rainfall.**

### 3.1.1 Groundwater

Figure 9 shows plausible evolution of groundwater levels for each circulation storyline. Given drier winters in clusters 1 and 2, groundwater levels were estimated to be normal to below normal across all boreholes by spring 2023. Wetter conditions over winter associated with circulation patterns in clusters 3 and 4 were estimated to lead to groundwater level recovery to above normal levels, particularly for boreholes in Lincolnshire (the more northerly catchments on the map) as groundwater levels at these relatively faster responding boreholes were already recovering after sufficient rainfall in autumn 2022. Similar to the pattern for some slow-responding river catchments, some boreholes in East Anglia (such as Washpit Farm and Old Hall Thurgaton) were still estimated to have a high likelihood of remaining at below normal levels by spring 2023 even with the wetter conditions from winters in clusters 3 and 4.

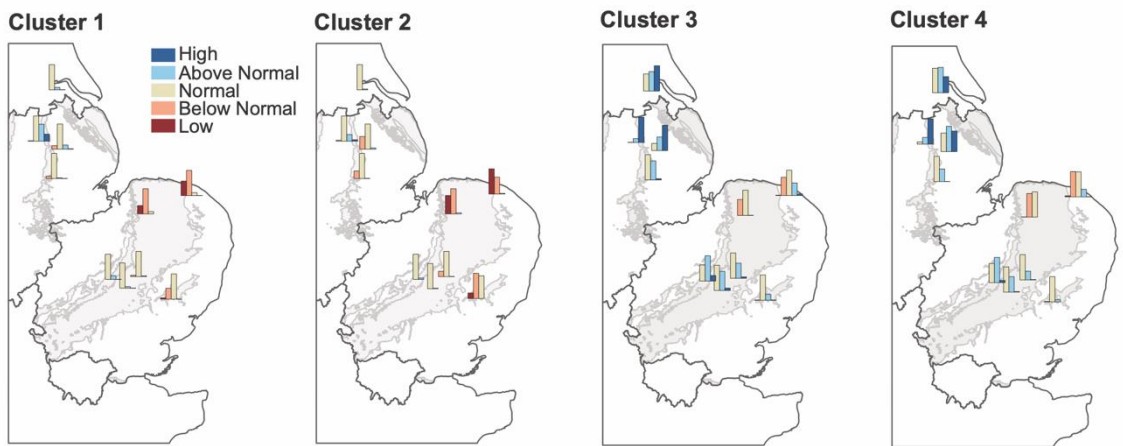

**Figure 9: Outlook of groundwater levels for each borehole in different categories (percentiles relative to 1965-2015) by spring 2023 for each storyline. Individual plots show the distribution of hindcast winters for each percentile category as indicated by the colour key in Figure 3. Grey shading shows major aquifers in eastern England from the hydrogeology map of the British Geological Survey (https://www.bgs.ac.uk/datasets/hydrogeology-625k/).**

## 3.2 Influence of spring and summer 2023

Figure 10 shows the influence of a second consecutive dry summer in 2023 on the development of severe drought conditions (SSI-3 < -1.5). An accumulation of three months is used here to provide an indication of the shorter-term seasonal effects. The effect of SSI accumulated over a longer time period at 12-months is shown in Figure S7. Slow responding catchments are more influenced by the effect of a dry winter in clusters 1 and 2 with a comparatively higher likelihood of reaching severe conditions.

A dry summer with 60% LTA rainfall (similar to summer 2018) results in a higher likelihood for severe drought conditions to develop, especially following a dry winter characterised by circulation patterns in clusters 1 and 2. Given the higher likelihood of a wetter than average winter in clusters 3 and 4, fewer catchments reach severe drought conditions if summer 2023 receives 100% LTA rainfall. For groundwater-dominated catchments, it is likely that severe drought conditions are reached even with 100% LTA rainfall in summer 2023 across all four storylines, even for clusters 3 and 4 with wetter than average winters. This

is reflected previously in Figure 6, showing that river flows were estimated to be unlikely to recover to normal levels for all four storylines.

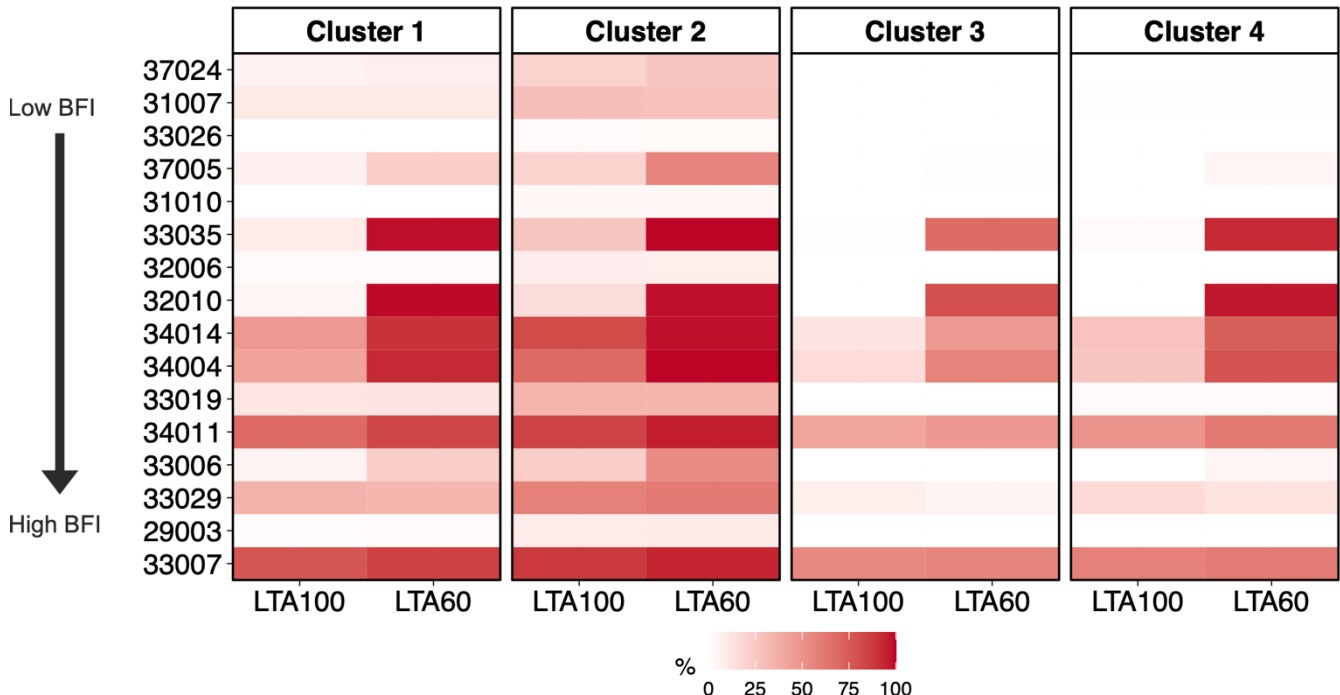

**Figure 10: Likelihood of reaching SSI-3 below -1.5 from winter 2022/23 to autumn 2023 for each circulation storyline. In the LTA100 experiment, spring to autumn 2023 are assumed to have rainfall at 100% long term seasonal average whereas the LTA60 experiment assumes summer 2023 to receive 60% LTA rainfall with 100% LTA rainfall for the other seasons. Catchments are ordered by increasing baseflow index (BFI) from the top.**

## 4 Discussion

Complementing hydrological outlooks with atmospheric circulation storylines adds a dynamical perspective to existing approaches to explore worst-case scenarios. The storylines cover a range of possible combinations of the various atmospheric circulation indices and span a wide range of surface rainfall and temperature responses. Outcomes across the circulation storylines encompass a wider range of outcomes compared to existing approaches which rely on the repetition of historical years. Considering storylines of the 2022 drought can increase risk awareness by enabling water managers to plan for water resources provisions if an upcoming season resembles certain atmospheric circulation patterns, and to explore plausible worst cases that are possibly outside the range of historical years. The widely used ensemble streamflow prediction (ESP) approach, assuming the repetition of past years, and the "possible worst case" scenarios used by the Environment Agency, representing synthetic rainfall time series at different percentages of long term average, can both be thought of as storylines describing possible ways an event might unfold, but often do not consider physical plausibility. The approach taken in this study supplements these existing products by sampling physically plausible rainfall occurrence modelled by SEAS5 with reference to their atmospheric drivers. Donegan et al. (2021) recently demonstrated the benefits of a NAO-conditioned ESP approach in planning for dry winters in Ireland by selecting historical years with information from hindcast prediction of the NAO. This is

also reflected in the results from this study which show the advantage of a conditioned approach with a more detailed focus on the drivers of rainfall in eastern England.

Although this study did not consider the likelihood of a particular storyline for winter 2022/23, further subsets to the storylines can be made over time or retrospectively. For example, storylines of winter 2022/23 could have been created by selecting only winters in the hindcasts with a wetter than average preceding November (as was observed in November 2022). When employed during an ongoing event, this approach may also shed light on the conditions required for drought termination, for example by calculating drought termination metrics in Parry et al. (2016) for each storyline. The same approach can be used after an event

to explore downward counterfactuals and the hydrological impacts should the event have turned out worse. For example, subsets of the storylines can be made to explore consequences should winter 2022/23 have turned out even drier than observed or if the preceding November had been drier than average.

      The results highlighted the conditions that could have led to continued drought conditions for catchments into 2023,

particularly in the northeast region of Anglian Water, due to a combination of insufficient winter rainfall and the effects of hydrogeology leading to long persistence of low flows. East Anglia received 69% of LTA rainfall in the observed winter 2022/23. The observed winter exhibited a NAO-/EA- pattern, resembling the atmospheric circulation patterns for winters in cluster 1. Similar to the composite mean SLP anomalies of cluster 1, winter 2022/23 saw high pressure conditions over the UK leading to drier than average conditions with the high pressure centre shifted further westwards (Figure 11). Results from

cluster 1 show that it was likely for flows to remain below normal by spring 2023 with the potential of continued drought conditions throughout 2023, particularly for groundwater-dominated catchments, even with spring to autumn 2023 receiving 100% LTA rainfall. The continued vulnerability of catchments in this region in 2023 resulted in a warning that water use restrictions could be needed and helped prioritise actions for the year ahead  (Anglian Water 2023). The likelihood of a return to drought conditions nationally was also reflected in the statement from the National Drought Group which stated that England

was "one hot, dry spell away from drought returning this summer" (National Drought Group 2023). The observed summer 2023 was wetter than average nationally, with a notably wet July. East Anglia received 95% of LTA rainfall with patches of slightly below average rainfall in parts of Norfolk. Parts of East Anglia remained in drought status through summer 2023 and official drought status was only lifted in October 2023 after river flows and ecological impacts recovered sufficiently.

It is important to emphasize that the storylines developed here were not meant to be forecasts of winter 2022/23 but instead represented hypotheses of possible river flow and drought responses to explore plausible worst cases. The approach is advantageous from a disaster risk reduction perspective as it increases risk awareness and enables water resources planners to consider a much wider diversity of plausible river flow trajectories. The storylines form one possible source of evidence during drought planning by providing a signal of potentially wet or dry outcomes to plan for. Drought storylines could help prioritise

and re-direct operational resources such as borehole maintenance in key areas (e.g. Norfolk) where the large sample of hindcast

winters show continued drought conditions or areas where plausible worst cases within each circulation storyline could exceed certain thresholds (e.g. relative to past reference droughts). As the large sample of pooled hindcasts cover a wide range of plausible outcomes, exploring storylines of an event in near real-time could also be beneficial for the practitioners (e.g. Environment Agency and environmental conservation organisations) as they can explore possible impacts to different sectors from either prolonged river flow deficits or abrupt drought-flood transitions. From a user's perspective, this approach is valuable as the skill of available forecasts, though continuously improving, is currently not perfect. Having the information on what a plausible worst-case might look like is therefore useful for planning purposes. However, in such an approach it is crucial to establish that the storylines are actually plausible, otherwise decision-making could be suboptimal. Yet doing so is inherently challenging since by definition, extreme storylines will lie in the tails of the distributions and will not be well sampled (if at all) in the observations. That is why we conducted model evaluation for our stated purpose, for precipitation (Figures 1 and 2), the circulation drivers (Figure 3), and the atmospheric patterns themselves (Figure 5). The reliability of the storyline approach ultimately rests on the physical plausibility of the storylines that are produced. However, this is the case for any exploration of extreme outcomes, and is not particular to the storyline approach.

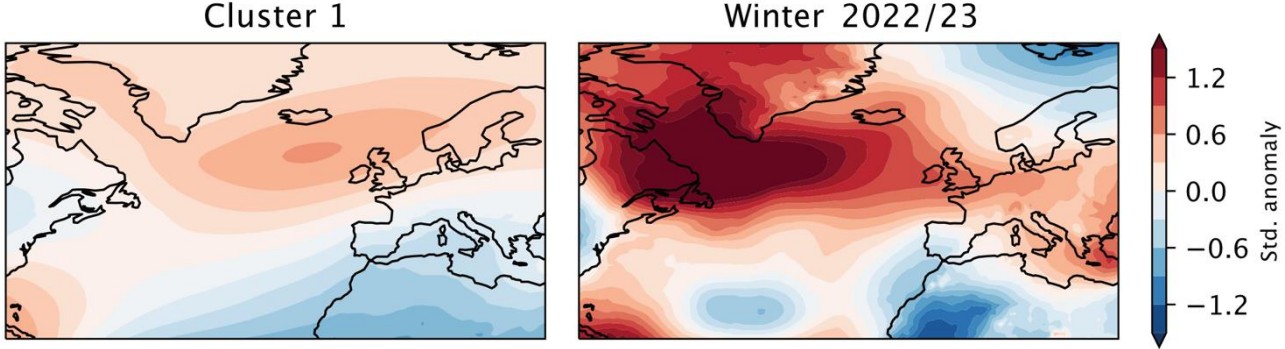

**Figure 11 Composite mean sea level pressure anomalies for winters in cluster 1 of the SEAS5 winters (left) compared to mean sea level pressure anomalies over observed winter 2022/23 (right: data from ERA5)**

This study contributes to the growing use of initialized model simulations to explore a large sample of plausible events to mitigate the challenge of short observational records and better consider internal climate variability (Kelder et al. 2022; Brunner and Slater 2022; Brunner et al. 2021; Chan et al. 2023). Although this study focused on the winter season given the importance of winter rainfall for the replenishment of river flows and aquifers in East Anglia, a similar approach can be taken for other seasons to provide complementary information. In this study, the NAO and EA patterns are influential for rainfall in East Anglia. For this approach to be applied in other regions, storylines should be conditioned on atmospheric circulation indices that are relevant to rainfall patterns in that particular region. This approach may be particularly useful when other forecasting approaches may be less informative and when it may be useful to consider a wider range of outcomes to explore

plausible worst cases (e.g. during a prolonged dry weather period prior to drought onset). For example, existing practice assumes the repetition of key individual years (such as the La Niña year of 2011) and results from this study highlight the benefits of considering a wider range of outcomes, including the combined effects of NAO and EA patterns during La Niña years. Operational drought forecasting tends to be done within water companies using calibrated hydrological models of key catchments and some companies already make use of seasonal forecasts. The level of resources required to extend existing methodologies to include a wider sample of seasonal hindcasts conditioned on atmospheric circulation patterns is minimal and could provide greater context with more robust evidence to inform the short to medium-term hydrological situation.

**5 Conclusions**

This study demonstrated the use of seasonal hindcasts over a historical period to create hydrological drought storylines conditioned on atmospheric circulation patterns. Using the 2022 drought as a case study, storylines of the 2022 drought for river flows and groundwater levels were created in autumn 2022 for East Anglia to represent the plausible progression over winter 2022/23 and spring-summer 2023 in order to explore the potential severity of the drought. Four circulation storylines were defined by clustering a large sample of winters in the SEAS5 hindcasts based on atmospheric circulation patterns responsible for rainfall anomalies in eastern England. Circulation storylines span the possible combinations of various atmospheric circulation indices and encompass a greater range of plausible outcomes compared to existing approaches. Results highlight the importance of winter rainfall, particularly for groundwater-dominated catchments, and the conditions that could have led to continued vulnerability for catchments and boreholes to severe drought conditions in 2023. The storylines show that hydrological drought conditions of the 2022 drought could have further intensified across most selected catchments, if winter 2022/23 had resembled a dry winter storyline, which actually transpired, and was subsequently followed by a second consecutive dry summer, which turned out to be wetter than average. This approach can be used in conjunction with existing methods in real time to plan for prolonged dry weather or ongoing droughts and explore plausible worst cases.

**Data availability**

Observed CEH-GEAR rainfall data is available from the Environmental Information Data Centre (EIDC) (https://doi.org/10.5285/dbf13dd5-90cd-457a-a986-f2f9dd97e93c - Tanguy et al. 2021). Observed HadUK-Grid temperature data is available from the Centre for Environmental Data Analysis (CEDA) archive (http://catalogue.ceda.ac.uk/uuid/4dc8450d889a491ebb20e724debe2dfb – Hollis et al. 2019). Daily SEAS5 hindcasts are available from Climate Data Store (CDS) (https://cds.climate.copernicus.eu/cdsapp - !/dataset/seasonal-original-single-levels). Data before 1993 can be requested using the CDS toolbox. ERA5 reanalysis data is available from CDS (https://cds.climate.copernicus.eu/cdsapp#!/dataset/reanalysis-era5-single-levels). Observed river flow data is obtained from

the National River Flow Archive (https://nrfa.ceh.ac.uk/). Simulated river flows and groundwater level from December 2022 to November 2023 for each storyline is hosted on the Zenodo archive (DOI: https://doi.org/10.5281/zenodo.7756582).

**Author contributions**

This study was completed during a PhD studentship placement at Anglian Water initiated by WCHC and GD. WCHC conducted the formal analysis and prepared the original paper. TGS, KF, GD, MT and NWA supervised the study. All authors contributed to the writing and interpretation of the results.

**Acknowledgements**

The authors would like to thank Santiago Rojas Arques (Anglian Water), Anne Bravery (Anglian Water) and Mark Sampson (Anglian Water) for their help in catchment/borehole selection, data access and the use of Aquimod to simulate groundwater levels at the selected boreholes. This research has been supported by the Natural Environment Research Council via the SCENARIO Doctoral Training Partnership (grant no. NE/S007261/1) and the NERC Climate Change in the Arctic-North Atlantic Region and Impacts on the UK (CANARI) project (grant no. NE/W004984/1).

**Competing interests**

The authors declare no competing interests

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
