# Peer review of "Added value of seasonal hindcasts to create UK hydrological drought storylines"

_Natural Hazards and Earth System Sciences, 2023_

## Author Comment (AC2)

**Added value of seasonal hindcasts for UK hydrological drought outlook**
Response to reviewers

**Reviewer #2**

Review of "Added value of seasonal hindcasts for UK hydrological drought" by Chan et al. This paper describes the use of storylines to assess worst-case scenarios of continuation of drought conditions into the next year, using 2022/2023 as an example. Meteorological hindcasts are from the SEAS5 system together with the GR6J hydrological model in the Greater Anglian region. The storylines uses clustering of the meteorological forcings using large-scale indices. The study highlights a useful tool to assess the risk of persisting drought over the winter period, and I recommend that it is published following a minor revision. However, I think the paper would eb strengthened by a more rigorous discussion on the skill and reliability of seasonal forecasts in Europe.

**Response:** We thank the reviewer for their positive comments about the manuscript. Please find below our responses to each of the comments.

Major comments

Uncertainty in seasonal forecasts. Seasonal forecasts are inherently uncertain, especially across Europe. The skill of the forecasts are not great on lead times longer than a month, and beyond that it is very questionable how much use the forecasts have. I think this needs to be discussed more in the paper

Verification. The authors could have used for example ERA5 reanalysis for verification of the seasonal forecasts. This could also have shed some light on the reliability of the forecasts, and through that have assessed the reliability and usefulness of the methodology

**Response:** Thank you for raising this point. We agree it is important to comment on the reliability of the SEAS5 hindcasts. This will be made much clearer in the revised version, also in response to comments from reviewer #3, as it appears that the objectives of our study were not well conveyed. However, we would like to point out that this paper is not about forecast verification and we are not attempting to predict the 2022 drought or assess the skill of the forecasts. Instead, we are using the SEAS5 hindcasts over all years, not just 2022, to explore "what-if" questions – i.e. how bad the 2022 drought could have been if followed by a dry winter characterized by a specific combination of atmospheric circulation indices. The reviewer is entirely correct in pointing out the limited skill of sub-seasonal and seasonal forecasts of drought, especially across Europe and particularly for the UK as shown by previous studies, such as Richardson et al. (2020). This underscores the very rationale behind our proposed methodology, which serves to complement traditional hydrological forecasting approaches. In light of the unreliability of forecasts, it becomes imperative

for water managers to grasp the potential implications of a plausible worst-case scenario to make informed decisions.

We propose to add the following figure to the revised manuscript. This follows the inclusion of similar figures in Thompson et al. (2017) and Kelder et al. (2020) to illustrate the reliability of large ensemble hindcast datasets (treated not as forecasts, but as plausible outcomes). Figure 1a shows observed total winter rainfall over the 1983-2016 period from the CEH-GEAR dataset (red dots) and the spread of modelled rainfall from the hindcasts across all ensemble members and lead times for all years over the same period (boxes). The observational data points fall within the box plots for roughly one-half of the years, and fall within the maximum and minimum in all years. Figure 1b shows the distribution of observed winter rainfall for each year standardised against the modelled distribution of winter rainfall for the same year. This shows that for 22 out of the 38 years, the observed value for a particular year is within 1 standard deviation of the modelled rainfall of that year. We hope the reviewer would agree that this figure helps show that the SEAS5 hindcasts are reliable and useful over the selected period. We will expand the methods section to outline the reliability of the SEAS5 hindcasts in the revised manuscript.

[Figure]

Figure 1 (left) Observed and simulated mean winter (DJF) rainfall over 1983-2016 from the SEAS5 hindcasts across 25 ensemble members and three lead times (Sept, Oct and Nov) over the Anglian region. The whiskers of the box plot extend to the maximum and minimum modelled value. (right) Distribution of observed winter rainfall for each year over 1983-2016 standardised against the distribution of simulated rainfall of the same year.

Minor comments

1. L31. You mention UK as the area here, but the study area is rather eastern England.

**Response:** We will make it clearer that although the study area is eastern England, the combination of NAO and EA influences winter rainfall for all regions of the UK and conditional outlooks can be applied to other regions.

2. The abstract is a bit too long and the methods is described quite convoluted. I would suggest to shorten this a bit.

**Response:** We will shorten the abstract. We believe the methods section is rightly longer for a paper like this given we are trying to set out a new framework to create drought storylines. We will try to make the methods section shorter but this might be challenging and might not be possible, especially in light of the later comments from the reviewer (i.e. comment 4 below) and related comments from reviewer #3 which would require elaborating on the use of the UNSEEN fidelity test and explaining more clearly the aims and objectives of our paper.

3. L42. SEAS5 provides forecasts up to 215 days, which is just over 7 months.

**Response:** We will correct this.

4. L76-78. The argument that there are 2850 winters in the hindcast dataset is a bit exaggerated since the ensemble members from each initialisation are not independent runs. Also, the model runs tend to approach the model cliamte over time. I think this can be further elaborated.

**Response:** Regarding the reviewer's point on drift towards model climatology, this is why we have tested for the independence and stability of the SEAS5 hindcasts across the three lead times in our paper (Lines 87-91 in the original manuscript). We will aim to make this clearer in the revised methods section. The use of initialised ensembles to explore plausible worst cases is well established, either by pooling climate model hindcasts as in Thompson et al. (2017) or by pooling weather forecasts as in Kelder et al. (2021) and Brunner and Slater (2022) among others. As detailed in Kelder et al. (2022), techniques have been developed to assess the model fidelity of seasonal hindcasts. First, the stability test recognizes that forecasts may drift towards the model climatology over time. This is tested by comparing the distribution of simulated winter rainfall between the three lead times. As seen from Fig 1b, the distribution of simulated winter rainfall is similar across the three lead times, indicating that there is no discernible drift. Second, the independence test assesses whether each ensemble member for each lead time is correlated with each other. This is done by considering the correlation between each pair of ensemble members over the 1982-2015 period for each lead time. As seen from Figure 1c in the manuscript, the median correlation for all three lead times is close to zero, thus showing that the ensemble members can be considered independent from each other over the considered timescale.

L99. The indices were calculated from MSLP. I would put that in the first sentence for clarity.

**Response:** We will correct.

5.  Figure 2. From the figure it looks like the EA has a clear correlation with rainfall anomaly, whereas NINO3.4 and NAO has very weak correlation with rainfall.

**Response:** We will make the suggested change in the text to describe that 1) there is no clear relationship between ENSO phase and rainfall anomalies, 2) there is a positive relationship between the EA index and rainfall anomalies, and 3) there is a weak negative relationship between the NAO index and rainfall anomalies.

6.  L184. The standard nowadays is to use KGE rather than NSE

**Response:** We thank the reviewer for this suggestion. In our paper, we have followed the multi-objective calibration strategy to determine the top parameter set for the GR6J hydrological model simulations. As introduced by Smith et al. (2019), the strategy was used successfully to calibrate hydrological models to reconstruct UK river flows and historic droughts. We have therefore not relied entirely on one statistical measure for model performance but instead have used a variety of model performance statistics (Nash-Sutcliffe Efficiency (NSE), NSE of logarithmic flows, absolute percentage bias, percentage error in Q95 (low flows) and percentage error in mean annual minimum (30-day averaged) flow) to take into account the full range of river flow characteristics. We chose to report only the logNSE values in Table 1 for brevity but we propose to add the model performance statistics for all statistical measures in a table in the supplementary materials in the revised manuscript.

7.  L199. Why do you state here that "Although winter 2022/23 has now passed.."? The approach is valid regardless if applied to previous cases or in real-time?

**Response:** The reviewer is correct that this approach is valid whether applied to a prior drought event or to an on-going event in real time. We will add a sentence to clarity this point. The retrospective discussion of winter 2022/23 mainly serves to show that there were still some concerns about water resources after winter 2022/23, and that the rainfall response similar to the winter that actually transpired is included within the larger sample of hindcast winters employed in this study.

8.  L214. What is slightly lower? How much of normal rainfall expressed in percentage?

**Response:** We will add the percentage of normal rainfall.

9.  L216. Yes, soil moisture would be depleted during a dry summer, but lower reservoirs would not be as much affected, which drought conditions are you discussing here?

**Response:** We will clarify the impacts of lower soil moisture on agricultural droughts.

10. Figure 5. The colour scale of the percentiles are not great, since normal conditions have a similar colour scheme as the below normal (yellow-orange-red). I would suggest to use a greenish colour to denote normal conditions.

**Response:** This particular colour scale is employed by the UK Hydrological Outlook (https://hydoutuk.net/), the UK Hydrological Summaries (https://nrfa.ceh.ac.uk/monthly-hydrological-summary-uk) and more recently the WMO HydroSOS (Status and Outlooks) portal (https://eip.ceh.ac.uk/hydrology/HydroSOS/portal/). The color scale has been tested in multiple iterations with a range of key stakeholders in workshops and focus groups in past projects aiming at better communicating hydro-meteorological information (e.g. AboutDrought, RADAR and EdGE). The colour scale takes into consideration factors such as colour-blind friendliness, user-friendliness and intuitive interpretation. Normal conditions are set to a tan colour in order to be relatively "unnoticable" so that outcomes that focus on extremes are more noticeable. We would prefer to keep the color scale as it is but propose to add thin borders to the bars to make them visually clearer.

**References**

Brunner, M. I. and Slater, L. J.: Extreme floods in Europe: going beyond observations using reforecast ensemble pooling, Hydrology and Earth System Sciences, 26, 469–482, https://doi.org/10.5194/hess-26-469-2022, 2022.

Kelder, T., Müller, M., Slater, L. J., Marjoribanks, T. I., Wilby, R. L., Prudhomme, C., Bohlinger, P., Ferranti, L., and Nipen, T.: Using UNSEEN trends to detect decadal changes in 100-year precipitation extremes, npj Clim Atmos Sci, 3, 1–13, https://doi.org/10.1038/s41612-020-00149-4, 2020.

Kelder, T., Marjoribanks, T. I., Slater, L. J., Prudhomme, C., Wilby, R. L., Wagemann, J., and Dunstone, N.: An open workflow to gain insights about low-likelihood high-impact weather events from initialized predictions, Meteorological Applications, 29, https://doi.org/10.1002/met.2065, 2022.

Richardson, D., Fowler, H. J., Kilsby, C. G., Neal, R., and Dankers, R.: Improving sub-seasonal forecast skill of meteorological drought: a weather pattern approach, Natural Hazards and Earth System Sciences, 20, 107–124, https://doi.org/10.5194/nhess-20-107-2020, 2020.

Smith, K. A., Barker, L. J., Tanguy, M., Parry, S., Harrigan, S., Legg, T. P., Prudhomme, C., and Hannaford, J.: A multi-objective ensemble approach to hydrological modelling in the UK: an application to historic drought reconstruction, Hydrology and Earth System Sciences, 23, 3247–3268, https://doi.org/10.5194/hess-23-3247-2019, 2019.

Thompson, V., Dunstone, N. J., Scaife, A. A., Smith, D. M., Slingo, J. M., Brown, S., and Belcher, S. E.: High risk of unprecedented UK rainfall in the current climate, Nature Communications, 8, 107, https://doi.org/10.1038/s41467-017-00275-3, 2017.

---

## Author Response (AR1)

**Added value of seasonal hindcasts for UK hydrological drought outlook**
Response to reviewers

**Editor's comments**

I'd like to thank the authors and reviewers for their efforts on preparation and reviewing of this manuscript. All reviews considered, I'd like to give authors the opportunity to submit a revised manuscript addressing all the comments. I highly encourage the authors to pay a closer attention to the major comments of reviewer 2 and 3. Particularly on the comments:

Reviewer 2: a more rigorous discussion on the skill and reliability of seasonal forecasts in Europe.

Reviewer 3: How can you justify justify the suitability of the storyline methodology, which is better suited for climate projections, for evaluating forecasts? As R3 pointed out assessing probabilistic forecasts - especially of extremes - on a single event is not the appropriate use of this method.

**Response:** We thank the editor for inviting a revision and for coordinating the review process. Regarding the comments from reviewers 2 and 3, we have made the following changes to the manuscript, which are further detailed in the individual response to each reviewer below. Line numbers refer to the manuscript version with tracked changes.

1. We have clarified the objectives of our study and made it clearer that our paper is not about forecast verification. We have also expanded on the reliability of the SEAS5 hindcasts over the study region and the selected period. This includes an additional figure (Figure 1 in revised manuscript).
2. We have clarified in the revised manuscript on the way we used the SEAS5 hindcasts. We hope the revised manuscript makes it clearer that we did not "assess probabilistic forecasts … on a single event". We have pooled the seasonal hindcasts and each hindcast winter was treated as an individual plausible outcome. Storylines of the 2022 drought were then created to represent how the 2022 drought could have unfolded if winter 2022/23 resembled the atmospheric circulation configuration of individual winters in the seasonal hindcast dataset.

**Reviewer #1**

Summary: The paper demonstrates the use of seasonal hindcasts for creating outlooks of ongoing droughts of river flows and groundwater levels conditioned on atmospheric circulation patterns. These outlooks can supplement traditional weather forecasts for exploring a wider range of plausible outcomes. The demonstration is made on the 2022 drought. Among others, the results of the case study indicate that

drier than average winters would result in the continuation of the drought with a high likelihood of below normal to low river flows by spring and summer.

General evaluation: Overall, I find that the paper is interesting, meaningful and very well-prepared with some room for improvement, which mainly concerns additional discussions that could be made. Specifically, I recommend minor revisions according to the comments provided here below.

**Response:** We thank the reviewer for their positive comments about the manuscript. We are glad to hear the reviewer finds the paper to be interesting. Our response to each of the minor comments is presented below (in red).

**Specific minor comments**

- Lines 63−64 state that "the Anglian region is particularly vulnerable to prolonged dry weather and droughts as it is the driest region in the UK". Maybe discussions could be added on whether and how the added value of the practical framework proposed is expected to deviate across regions and across climatic conditions.

**Response:** We agree that there could be more discussion on the applicability of this approach across different regions. We have made this clearer in the revised discussion section to emphasize that this approach can be applied across different regions and climatic conditions, but storylines created for other regions should be conditioned on various combinations of circulation indices that are relevant for rainfall patterns in that region. For this study, the EA pattern is influential for rainfall in the Anglian region, hence there is added value of incorporating seasonal hindcasts of the EA pattern in hydrological outlooks to understand plausible worst cases (Lines 411-413 in revised manuscript).

- Lines 305−318 discuss the value of the proposed framework by comparing its output for the case examined with the observed winter 2022/23. In my view, these discussions are important for the paper and maybe they could also be supported by one or more figures. For instance, Figure S7 could be moved in the main manuscript.

**Response:** The retrospective discussion of winter 2022/23 serves to show that there were still some concerns about water resources after winter 2022/23, and that a rainfall response similar to the winter that actually transpired is included within the larger sample of hindcast winters employed in this study. On the latter point, we have, as suggested, moved Figure S7 to the main text (Figure 11 in the revised manuscript) as it shows that the circulation patterns observed for winter 2022/23 resemble winters in Cluster 1 of the hindcast winters.

It should be noted that we are not attempting to predict or forecast winter 2022/23. Instead, conditional storylines explore plausible worst cases that are outside the range of historical years and enable water managers to prioritise or re-direct operational resources to prepare for such an outcome. Thus, we do not think that extensive discussion of the observed winter is required or warranted. We have made the aims of our study clearer in the abstract (L20-26), introduction (L89-92 and L95-99) and methods section (L106-108).

- Right after lines 305−318, extensive discussions (and maybe a figure as well) could be added to highlight the benefits of applying the new framework along with the use of traditional weather forecasts, instead of using such forecasts only, in the case examined.

**Response:** We have expanded on our discussion of the use of conditional storylines alongside traditional weather forecasts. We have included an additional figure (Figure 7 in the revised manuscript) comparing the widely used ensemble streamflow prediction (ESP) technique with the results from each circulation storyline which shows the use of a larger sample of weather sequences from pooling seasonal hindcasts can enhance risk awareness as it encompasses a wider range of outcomes compared to the traditional ESP technique which relies on repeating historical years (L289-294).

We have also further discussed the utility of conditional storylines to explore plausible worst cases in the revised discussion – i.e. how bad could the 2022 drought have been given a winter with specific atmospheric circulation patterns. Understanding plausible worst cases of the hydrological drought in advance is beneficial for water companies as they can prepare for both supply-side measures to increase abstraction or demand-side measures to increase water efficiency. As the large sample of pooled hindcasts cover a wide range of plausible outcomes, exploring storylines of an event in near real-time could also be beneficial for the environmental regulator (e.g. Environment Agency and environmental conservation organisations) as they can explore possible impacts to the natural environment from either prolonged river flow deficits or abrupt drought-flood transitions. From a user's (e.g. water resources manager) perspective, this approach is valuable as the skill of available forecasts, though continuously improving, are currently not perfect. Having the information on what a plausible worst-case might look like is therefore useful from a water management perspective for planning purposes. The storylines created in this study also adds a dynamical perspective to existing approaches and provides a connection to physical climate drivers. This aids physical understanding and may be used to track the evolution of drivers and its impacts as the season progresses. We have highlighted these advantages in the discussion section (Lines 393-401 in revised manuscript).

**Reviewer #2**

Review of "Added value of seasonal hindcasts for UK hydrological drought" by Chan et al. This paper describes the use of storylines to assess worst-case scenarios of continuation of drought conditions into the next year, using 2022/2023 as an example. Meteorological hindcasts are from the SEAS5 system together with the GR6J hydrological model in the Greater Anglian region. The storylines uses clustering of the meteorological forcings using large-scale indices. The study highlights a useful tool to assess the risk of persisting drought over the winter period, and I recommend that it is published following a minor revision. However, I think the paper would eb strengthened by a more rigorous discussion on the skill and reliability of seasonal forecasts in Europe.

**Response:** We thank the reviewer for their positive comments about the manuscript. Please find below our responses to each of the comments.

Major comments

Uncertainty in seasonal forecasts. Seasonal forecasts are inherently uncertain, especially across Europe. The skill of the forecasts are not great on lead times longer than a month, and beyond that it is very questionable how much use the forecasts have. I think this needs to be discussed more in the paper

Verification. The authors could have used for example ERA5 reanalysis for verification of the seasonal forecasts. This could also have shed some light on the reliability of the forecasts, and through that have assessed the reliability and usefulness of the methodology

**Response:** Thank you for raising this point. We agree it is important to comment on the reliability of the SEAS5 hindcasts. We have made this clearer in the revised version, also in response to comments from reviewer #3, as it appears that the objectives of our study were not well conveyed. However, we would like to point out that this paper is not about forecast verification and we are not attempting to predict the 2022 drought or assess the skill of the forecasts. Instead, we are using the SEAS5 hindcasts over all years, not just 2022, to explore "what-if" questions – i.e. how bad the 2022 drought could have been if followed by a dry winter characterized by a specific combination of atmospheric circulation indices. The reviewer is entirely correct in pointing out the limited skill of sub-seasonal and seasonal forecasts of drought. We have added a comment on this, referencing past work showing limited skill especially across Europe and particularly for the UK (Richardson et al. 2020) (Lines 51-53). This underscores the very rationale behind our proposed methodology, which serves to complement traditional hydrological forecasting approaches. In light of the unreliability of forecasts, it becomes imperative for water managers to grasp the potential implications of a plausible worst-case scenario to make informed decisions.

We have added the following figure and additional explanation to the revised manuscript (Lines 108-116). This follows the inclusion of similar figures in Thompson et al. (2017) and Kelder et al. (2020) to illustrate the reliability of large ensemble hindcast datasets (treated not as forecasts, but as plausible outcomes). Figure 1a shows observed total winter rainfall over the 1983-2016 period from the CEH-GEAR dataset (red dots) and the spread of modelled rainfall from the hindcasts across all ensemble members and lead times for all years over the same period (boxes). The observational data points fall within the box plots for roughly one-half of the years, and fall within the maximum and minimum in all years. Figure 1b shows the distribution of observed winter rainfall for each year standardised against the modelled distribution of winter rainfall for the same year. This shows that for 22 out of the 38 years, the observed value for a particular year is within 1 standard deviation of the modelled rainfall of that year. We hope the reviewer would agree that this figure and the expanded explanation helps show that the SEAS5 hindcasts are reliable and useful over the selected period.

[Figure]

Figure 1 (left) Observed and simulated mean winter (DJF) rainfall over 1983-2016 from the SEAS5 hindcasts across 25 ensemble members and three lead times (Sept, Oct and Nov) over the Anglian region. (right) Distribution of observed winter rainfall for each year over 1983-2016 standardised against the distribution of simulated rainfall of the same year. The whiskers of the box plot extend to the maximum and minimum modelled value.

Minor comments

1. L31. You mention UK as the area here, but the study area is rather eastern England.

**Response:** We have made it clearer in the discussion section that although the study area is eastern England, the combination of NAO and EA influences winter rainfall for

all regions of the UK and conditional outlooks can be applied to other regions (L419-422).

2. The abstract is a bit too long and the methods is described quite convoluted. I would suggest to shorten this a bit.

**Response:** We have shortened the abstract. We believe the methods section is rightly longer for a paper like this given we are trying to set out a new framework to create drought storylines, especially in light of the later comments from the reviewer (i.e. comment 4 below) and related comments from reviewer #3 which have required elaborating on the use of the UNSEEN fidelity test and explaining more clearly the aims and objectives of our paper.

3. L42. SEAS5 provides forecasts up to 215 days, which is just over 7 months.

**Response:** We have corrected this.

4. L76-78. The argument that there are 2850 winters in the hindcast dataset is a bit exaggerated since the ensemble members from each initialisation are not independent runs. Also, the model runs tend to approach the model cliamte over time. I think this can be further elaborated.

**Response:** Regarding the reviewer's point on drift towards model climatology, this is why we have tested for the independence and stability of the SEAS5 hindcasts across the three lead times in our paper (Lines 87-91 in the original manuscript). We have made this clearer in the revised methods section. The use of initialised ensembles to explore plausible worst cases is well established, either by pooling climate model hindcasts as in Thompson et al. (2017), or by pooling weather forecasts as in Kelder et al. (2021) and Brunner and Slater (2022) among others. We have added these references in the methods section (L108-110 in the revised manuscript). As detailed in Kelder et al. (2022), techniques have been developed to assess the model fidelity of seasonal hindcasts. First, the stability test recognizes that forecasts may drift towards the model climatology over time. This is tested by comparing the distribution of simulated winter rainfall between the three lead times. As seen from Fig 2b, the distribution of simulated winter rainfall is similar across the three lead times, indicating that there is no discernible drift. Second, the independence test assesses whether each ensemble member for each lead time is correlated with each other. This is done by considering the correlation between each pair of ensemble members over the 1982-2015 period for each lead time. As seen from Figure 2c in the manuscript, the median correlation for all three lead times is close to zero, thus showing that the ensemble members can be considered independent from each other over the considered timescale. We have added the information above in the methods section (L132-137 in the revised manuscript).

L99. The indices were calculated from MSLP. I would put that in the first sentence for clarity.

**Response:** We have corrected this (L147-148 in revised manuscript).

5. Figure 2. From the figure it looks like the EA has a clear correlation with rainfall anomaly, whereas NINO3.4 and NAO has very weak correlation with rainfall.

**Response:** We have made the suggested change in the text to describe that 1) there is no clear relationship between ENSO phase and rainfall anomalies, 2) there is a positive relationship between the EA index and rainfall anomalies, and 3) there is a weak negative relationship between the NAO index and rainfall anomalies (Lines 155-156). Although previous studies such as Folland et al. (2015) suggested an association of La Niña with dry winters, the relationship between ENSO phase and winter rainfall is weak and is influenced by other drivers of rainfall. The choice to select for La Niña winters in the hindcasts was motivated by the fact that La Niña conditions was observed over 2022.

6. L184. The standard nowadays is to use KGE rather than NSE

**Response:** We thank the reviewer for this suggestion. In our paper, we have followed the multi-objective calibration strategy to determine the top parameter set for the GR6J hydrological model simulations. As introduced by Smith et al. (2019), the strategy was used successfully to calibrate hydrological models to reconstruct UK river flows and historic droughts. We have therefore not relied entirely on one statistical measure for model performance but instead have used a variety of model performance statistics (Nash-Sutcliffe Efficiency (NSE), NSE of logarithmic flows, absolute percentage bias, percentage error in Q95 (low flows) and percentage error in mean annual minimum (30-day averaged) flow) to take into account the full range of river flow characteristics. We chose to report only the logNSE values in Table 1 for brevity but we have added a figure in the supplementary materials to show the model performance statistics for all statistical measures (Figure S3 in the revised manuscript).

7. L199. Why do you state here that "Although winter 2022/23 has now passed.."? The approach is valid regardless if applied to previous cases or in real-time?

**Response:** The reviewer is correct that this approach is valid whether applied to a prior drought event or to an on-going event in real time. The retrospective discussion of winter 2022/23 mainly serves to show that there were still some concerns about water resources after winter 2022/23, and that a rainfall response similar to the winter that actually transpired is included within the larger sample of hindcast winters employed in this study.

8. L214. What is slightly lower? How much of normal rainfall expressed in percentage?

**Response:** We have added the percentage of normal rainfall.

9. L216. Yes, soil moisture would be depleted during a dry summer, but lower reservoirs would not be as much affected, which drought conditions are you discussing here?

**Response:** We have clarified the impacts of lower soil moisture on agricultural droughts.

10. Figure 5. The colour scale of the percentiles are not great, since normal conditions have a similar colour scheme as the below normal (yellow-orange-red). I would suggest to use a greenish colour to denote normal conditions.

**Response:** This particular colour scale is employed by the UK Hydrological Outlook (https://hydoutuk.net/), the UK Hydrological Summaries (https://nrfa.ceh.ac.uk/monthly-hydrological-summary-uk) and more recently the WMO HydroSOS (Status and Outlooks) portal (https://eip.ceh.ac.uk/hydrology/HydroSOS/portal/). The colour scale has been tested in multiple iterations with a range of key stakeholders in workshops and focus groups in past projects aiming at better communicating hydro-meteorological information (e.g. AboutDrought, RADAR and EdGE). The colour scale takes into consideration factors such as colour-blind friendliness, user-friendliness and intuitive interpretation. Normal conditions are set to a tan colour in order to be relatively "unnoticable" so that outcomes that focus on extremes are more noticeable. We have kept the colour scale as it is but added thin borders to the bars to make them visually clearer (Figures 6 and 9 in the revised manuscript and Figure S6 in the revised supplementary materials).

**Reviewer #3**

This study assesses ECMWF SYS5 for use in drought outlooks over the UK. The paper is beautifully written, the figures are nicely presented and I could clearly follow at every point what the authors did. I really regret to say, then, that I thought the paper was ultimately misguided and that I do not recommend the paper to be published, for the following reasons:

1) The paper makes use of methods from climate studies (notably assessments of climatological distributions from the UNSEEN project) that are inadequate for ensemble forecast verification. As I note in the specific comments below, the accuracy and skill of forecasts can be assessed directly using well-established forecast verification methods, as can the appropriateness of ensemble spread,

which consider the correlation of forecasts with observations. I recommend the authors familiarise themselves with the fundamentals of forecast verification (see, e.g., Joliffe & Stephenson 2011) before reconfiguring their paper.

2) To me, the use of story lines is fundamentally at odds with the aims of ensemble forecasting. The conception of story lines is appropriate in climate projections, where ensembles of different GCMs are not formally statistically exchangeable, and thus should not be used to express, for example, quantitative confidence intervals. Story lines in climate change projections can be thought of as hypotheses. In ensemble forecasting, however, ensemble members should be formally exchangeable (i.e., each member is equally probable), and we can directly test the appropriateness of the ensemble following (1). This means ensembles can be used to assign probabilities to events. Developing story lines based on climate drivers essentially does away with this probabilistic information in preference to a narrative-drive prediction method. The purpose of ensemble weather and climate forecasting can be thought of as an attempt to get away from narrative-style predictions: basically, the uncertainty in weather is irreducible and cannot be distilled to one or two 'story lines'. Because weather is chaotic, when ensembles are constructed correctly, they should be on average more accurate than any single-value forecast.

3) Following on from (2), it's not appropriate to assess probabilistic forecasts - especially of extremes - on a single event as is done in this paper. Probabilistic forecasts must be assessed on a population of events, and the population must be unbiased. Selecting such a population on the basis of when an extreme event (e.g., a drought) is observed isn't correct: it produces a biased population. The reasons for this are both intuitive and highly technical - e.g., it's not possible to assess false alarms when an extreme event is always observed in your population; for more technical reasons see Lerch et al. (2017) and Bellier et al. (2017)

4) Given (1) and (2) , it's not appropriate to assess the usefulness of ECMWF SYS5 for drought prediction solely on its ability to replicate a climatology; nor to use it to characterise climate drivers of rainfall. If the authors wanted to reconfigure their paper, reusing some of their techniques to identify climate drivers, the authors might consider:

   i) Comparing whether observed teleconnections (as seen in, e.g., reanalyses) are reproduced in SYS5, and whether this changes with lead time

   ii) Whether drivers of teleconnections are well forecast by SYS5 (following (3), when the forecasting system predicts such an event, rather than if one is observed)

   iii) Using (i) and (ii) to address questions on the conditional skill of SYS5 forecasts. For example, it could be used to answer questions such as: a) how well does SYS5

predict key drivers of drought? (b) how well does it do at reproducing observed teleconnections and (c) how do (a) and (b) relate to forecast skill?

These are merely suggestions of course; I leave it to the authors as to what they may wish to pursue. In any case, if the authors did follow these recommendations, I think it would be a fundamentally different paper to what is presented, and hence my recommendation to reject (rather than revise the paper).

It's always difficult to reject a paper like this, in which the authors show a good command of statistical and climate analyses, not to mention clear scientific writing and presentation, but which is in other ways seriously flawed (at least in my view). I really hope the authors don't find my review too discouraging - I wish them well reconfiguring their work and analyses to better align with the precepts of forecast verification.

**Response:** We would like to thank reviewer #3 for their detailed review of our paper. We are disappointed that the reviewer recommended a rejection of our paper. We note that similar concerns were not expressed by reviewers #1 and #2, and both reviews were positive with minor changes requested. However, reviewer #3's comments have brought to our attention that the presentation of the aims and objectives of our paper were clearly deficient and we believe this has led to a fundamental misunderstanding of the aims of our study. We apologise for the confusion generated. In a revised version, we have added to the introduction to clearly explain the objectives of the work and throughout the manuscript to emphasize our objectives and, more importantly, what our paper is not aiming to do.

1) This paper is not about forecast verification. We are not attempting to predict the 2022 drought event or assess the skill of the SEAS5 hindcasts in predicting the drivers of a particular drought year. Instead, the aim of this paper is to explore "what-if" situations of the 2022 drought should winter 2022/23 resemble specific atmospheric circulation patterns. We do this by pooling a large sample of hindcast data and clustering them using various circulation indices known to drive winter rainfall in the UK to create circulation storylines. We believe our use of the word "outlook", particularly in the paper title, could have contributed to the confusion that this paper is about forecast verification. We have amended the title to "Added value of seasonal hindcasts for UK hydrological drought storylines". The drought storylines created allow users to explore in real-time what the plausible worst case could be during an ongoing drought event, hence can be considered a worst-case outlook of the drought event, and not an outlook in the sense of a forecast. We have made it clear throughout the manuscript that our study is not attempting to predict the

outcome of the 2022 drought and have amended any potentially confusing wording to reflect this.

2) In relation to point 2 raised by the reviewer, we disagree that the use of storylines in this way is "at odds with the aims of ensemble forecasting". We did not advocate for the replacement of ensemble weather forecasting with storylines. The reviewer is correct that storylines can be thought of as hypotheses. The hindcast winters are treated as individual plausible outcomes and we have attached no probabilities to each storyline. Appending the circulation storylines in place of winter 2022/23 therefore enables the exploration of plausible worst cases – e.g. how bad could the 2022 drought have been if followed by a dry winter characterized by a specific combination of atmospheric circulation indices. As outlined in our response to reviewer #1, we have added that from a user's (e.g. water resources manager) perspective, this approach is valuable as the skill of available forecasts, though continuously improving, is currently not perfect. Having the information on what a plausible worst-case might look like is therefore useful from a water management perspective for planning purposes (L404-409) in the revised manuscript). For example, during a period of prolonged dry weather, conditional storylines can be created several months/seasons ahead to explore the potential range of outcomes should upcoming seasons resemble specific atmospheric circulation patterns with a particular focus on potential impacts of worst cases, in order to plan accordingly. While there have been advances in probabilistic forecasts, plausible worst cases will by definition lie in the tail of the distribution and their likelihood will not be well represented by finite-sized ensembles. Given the irreducibility of uncertainties in weather as the reviewer correctly identifies, we believe that understanding plausible worst cases during an event can be valuable as a "perfect" probabilistic forecast may not be attainable (added in L257-260 and L404-409 in the revised manuscript).

This paper argues that the use of conditional storylines could be a complementary tool (hence the "Added value" in our title) to ensemble forecasting. The well-established ensemble streamflow prediction (ESP) technique, where historical years are repeated, can also be thought of as following a storyline approach (i.e. what if the rainfall sequence from a historical year is repeated). ESP was first developed in the U.S. and is now a widely used approach worldwide for operational river flow prediction (Twedt et al. 1977; Day et al. 1985). This paper extends this methodology by adding information on atmospheric circulation drivers of rainfall and by exploring a wider range of outcomes. Additionally, the UK Environment Agency, the public body tasked with overseeing water abstraction licences, managing water transfers and preserving environmental flows, already uses storylines in their operational water resources management, especially during on-going droughts, where hydrological models are driven to produce 'possible worstcase' with hypothetical synthetic rainfall time series, which are simply % of long-term average, but with no consideration of how physically plausible that rainfall occurrence is. The approach proposed in our study is a clear improvement to this, as we sample physically plausible rainfall occurrence modelled by SEAS5 (as reflected in L63-69 in the Introduction and L367-371 in the Discussion). As noted in our response to reviewer #1, we have expanded on our discussion of how storylines can complement traditional ensemble forecasting approaches, and to include an additional figure comparing results from a traditional ESP framework and the circulation storylines (added as Figure 7 in the revised manuscript and explanation in L289-294).

3) In response to point 3 raised by the reviewer, we did not "assess probabilistic forecasts ... on a single event". As noted previously, we pooled SEAS5 hindcasts and clustered them according to known drivers of winter rainfall in our study region. We thus end up with a large sample of plausible winters separated by rainfall response from different combinations of circulation patterns. Contrary to the reviewer's concern, an extreme event is not always observed in the SEAS5 hindcast population – the outlooks clearly show that there are winters that are wet enough to have abruptly terminated the 2022 drought across the Anglian region.

4) We thank the reviewer for providing suggestions for research ideas outlined in point 4 - they are all interesting topics worthy of future research. However, the aims of these research ideas are fundamentally different to the aims of our paper. We believe our paper outlines a novel approach in relation to the use of meteorological information to aid decision-making by water resources managers and enhance risk awareness during a drought.

Specific comments

P3 L78-79 "Each ensemble member is perturbed with different initial atmospheric and ocean initial conditions" On first reading this I though this wording implied that the authors are (re) perturbing ensemble members, which I would think is unlikely given the computational demands of SYS5 and the authors' earlier declaration that they are using the retrospective forecast dataset. I assume the authors mean something like: "SYS5 ensemble members are generated by perturbing initial conditions", so if I'm right I suggest the authors go with something like this. In addition, it's my understanding ECMWF perturbs model physics in SYS5, which the authors might also want to mention.

**Response:** We have corrected as suggested – L104-105

P4 Section 2.1. I found this method of forecast verification basically inappropriate, as follows:

1) Using the UNSEEN framework isn't really appropriate here: that paper tried to put a flood event in climatological context, therefore assembling a large ensemble to assess describe the climatology makes sense. But we are dealing with forecasts here, which are expected to be correlated with observations. Assessing a simple model climatology is not enough to demonstrate the value of forecasts.

2) Even given (1), the idea that the model performs well at simulating a climatology isn't well demonstrated by Figure 1. For example, the figure shows that variance of winter rainfall is understated by SYS5. Further, variance can change with lead time, as information from initial conditions wanes and the model reverts to its internal climatology. Finally, it is quite possible to demonstrate bias etc. across multiple sites and lead times, rather than restricting it to a single site and pooling lead times, which occludes valuable information about the utility of the forecasts.

3) As noted in (1) Forecasts are expected to be correlated with observations. This means that the utility of forecasts is usually measured by:

   1) Forecast skill (i.e., forecast errors with respect to a climatological forecast, computed with appropriate error scores such as the continuous ranked probability score), conditioned on both location and lead time.

   2) The reliability of ensemble spread, using appropriate measures such as probability integral transforms, attributes diagrams or spread-error diagrams.

   In addition, useful information about forecasts is also:

   a) The sharpness of the forecasts

   b) ability to replicate observed climatology - e.g., with bias/variance/etc. It is only this last criterion that is assessed in the paper.

**Response:**

1) In response to high-level points #1 and #2 in the specific comments, we reiterate that this paper is not about forecast verification; its aim is not to assess whether the SEAS5 forecasts accurately specific drought events. We have improved the clarity of the explanation regarding the purpose of the UNSEEN model fidelity tests, as it appears that this aspect has not been entirely clear and seems to have led to some misunderstanding. The UNSEEN fidelity test introduced in Thompson et al. (2017) aims to assess whether the model data can be considered as alternative realizations of the real world. Observed winter rainfall from the historical period is just one realisation out of many possible alternative realisations that could have happened. As there is only one

observed value per year, the purpose of the UNSEEN framework is to consider the spread of simulated winter rainfall in the SEAS5 hindcasts compared to the observations. If the observed sample statistic falls within 95% of the modelled distribution, the modelled rainfall is considered to be statistically indistinguishable from the observations, as seen in Figure 1a in the original manuscript (Figure 2 in the revised manuscript). This follows the now well-established use of initialised ensembles where most of the initial-condition skill has been lost, either by pooling climate model hindcasts as in Thompson et al. (2017) or by pooling seasonal hindcasts as in van den Brink et al. (2004), Kelder et al. (2020) and Brunner and Slater (2022) among others. Individual forecasts are not discernibly correlated (as is mentioned in the next paragraph), and the different lead times are not discernibly dependent, so there is little skill in these forecasts. That is why we believe our use of the SEAS5 hindcasts, considered over all years rather than just for single years, as plausible realisations of winter weather, is appropriate. We have added these references to the revised manuscript and expanded on our justification of why the SEAS5 hindcasts can be considered reliable for our use (L108-116 in the revised manuscript).

2) In relation to point #3 in the specific comments, the reviewer raised the concern that forecasts are expected to be correlated with the observations. This does not detract from the results of this study. As our study is not about forecast verification, the wide range of plausible outcomes from pooling the seasonal hindcasts is important given our aims of exploring "what-if" situations and plausible worst cases. The reviewer's point on model climatology is valid, which is why we had tested for the independence and stability of the SEAS5 hindcasts in representing observed climatology across the three lead times. As detailed in Kelder et al. (2022), techniques have been developed to assess the model fidelity of seasonal hindcasts. First, the stability test recognizes that forecasts may drift towards the model climatology over time. This is tested by a comparison of the distribution of simulated winter rainfall between the three lead times. As seen from Figure 1b of the original manuscript (Figure 2 in the revised manuscript), the distribution of simulated winter rainfall is similar across the three lead times, indicating that there is no drift within this time scale. Second, the independence test assesses whether each ensemble member for each lead time is correlated with each other. This is done by considering the correlation between each pair of ensemble members over the 1982-2015 period for each lead time. As seen from Figure 2c in the revised manuscript, the median correlation for all three lead times is close to zero, thus showing that the ensemble members can be considered independent from each other. We have added further explanation of this in the revised manuscript (L132-137 in the revised manuscript).

**References**

Brunner, M. I. and Slater, L. J.: Extreme floods in Europe: going beyond observations using reforecast ensemble pooling, Hydrology and Earth System Sciences, 26, 469–482, https://doi.org/10.5194/hess-26-469-2022, 2022.

Day, G. N.: Extended Streamflow Forecasting Using NWSRFS, J. Water Resour. Plan. Manag., 111, 642–654, 1985.

Kelder, T., Müller, M., Slater, L. J., Marjoribanks, T. I., Wilby, R. L., Prudhomme, C., Bohlinger, P., Ferranti, L., and Nipen, T.: Using UNSEEN trends to detect decadal changes in 100-year precipitation extremes, npj Clim Atmos Sci, 3, 1–13, https://doi.org/10.1038/s41612-020-00149-4, 2020.

Kelder, T., Marjoribanks, T. I., Slater, L. J., Prudhomme, C., Wilby, R. L., Wagemann, J., and Dunstone, N.: An open workflow to gain insights about low-likelihood high-impact weather events from initialized predictions, Meteorological Applications, 29, https://doi.org/10.1002/met.2065, 2022.

Richardson, D., Fowler, H. J., Kilsby, C. G., Neal, R., and Dankers, R.: Improving sub-seasonal forecast skill of meteorological drought: a weather pattern approach, Natural Hazards and Earth System Sciences, 20, 107–124, https://doi.org/10.5194/nhess-20-107-2020, 2020.

Smith, K. A., Barker, L. J., Tanguy, M., Parry, S., Harrigan, S., Legg, T. P., Prudhomme, C., and Hannaford, J.: A multi-objective ensemble approach to hydrological modelling in the UK: an application to historic drought reconstruction, Hydrology and Earth System Sciences, 23, 3247–3268, https://doi.org/10.5194/hess-23-3247-2019, 2019.

Thompson, V., Dunstone, N. J., Scaife, A. A., Smith, D. M., Slingo, J. M., Brown, S., and Belcher, S. E.: High risk of unprecedented UK rainfall in the current climate, Nature Communications, 8, 107, https://doi.org/10.1038/s41467-017-00275-3, 2017.

Twedt, T. M., Schaake Jr., J. C., and Peck, E. L.: National Weather Service extended streamflow prediction, in Proceedings of the 45th Annual Western Snow Conference, 52–57, Albuquerque, New Mexico, available at: https://westernsnowconference.org/node/1106

---

## Author Response (AR2)

**Added value of seasonal hindcasts to create UK hydrological drought storylines**
Response to reviewers

**Editor's comments**

Thank you for your effort in revising the manuscript. Both reviewers are happy with the revisions. That said, one of the reviewers recommended "accepted subject to minor revisions". I read their comment, and I'd like to ask the authors to revise their manuscript one more time. This reviewer is a senior scientist and practitioner working closely with governments and decision makers. It'd a missed opportunity not to include their comments. I think the suggested minor revision put your paper in a practice-informed context that benefits you as authors as well as future readers. I'll handle the next round myself to expedite the review. Thanks for your patience and well done with the process all the way to here. Just the last notch.

**Response:** We thank the editor for handling the manuscript and recommending minor revisions. We appreciate the opportunity to amend the paper to enhance relevance to practitioners. Please find below our response to reviewer 2 (in red) and additions made to the revised manuscript to address the reviewer's two remaining concerns.

**Reviewer #2**

I appreciate the considerable efforts the authors have made to clarify that their methods are not designed as forecasts nor intended to be used as forecasts. This removes much of my original objections to the paper (with a some specific exceptions, detailed below). The strengths of the paper remain: it is very well written, the methods are clearly described, the analyses are appropriate and figures are well-presented.

**Response:** We would like to thank Reviewer #2 for reviewing the revised version of the manuscript. Following the valuable suggestions from the reviewer, we are glad that many of the original objections the reviewer had have been satisfactorily addressed after the first round of revisions.

I remain skeptical of the utility of this work for informing decisions. What the authors have effectively done is expand a climatological distribution by considering an ensemble of Seasonal forecasts as 'plausible' realisations. This may or may not be a a reasonable thing to do: we know, for example, that seasonal forecasts, including their dynamical representations of the world, become increasingly less accurate and less realistic at longer lead times (as they become more distant from initial conditions). One risk of the methods presented in this paper is that they artificially inflate the variance of a climatological distribution, leading to unrealistically catastrophic events (either wet or dry) in their storylines at the tails of the distributions. This can lead to overly conservative (and thus suboptimal) decision

making. I think the authors should acknowledge this possibility somewhere in their paper (can be as brief as a single sentence). As I noted in the previous review, forecasts are most useful when they are as sharp as possible without being over-confident: this narrows the range of possibilities for decision makers (usually explicitly in comparison to climatology). Widening this range of possibilities - perhaps artificially - may well make decision making less optimal. Having said all this, I accept that these are somewhat philosophical objections, and others can judge the work once it is published. As the methods are very clearly described, others can readily decide whether this approach is useful.

**Response:** If our understanding is correct, the reviewer seems to be addressing two distinct points here. The first is the fact that storylines are not sharp forecasts. Since it is current operational practice in water resources management to consider worst-case scenarios (which is effectively a storyline approach as we have further discussed after the first round of revisions) for the purpose of stress-testing, we believe that our approach is valuable for decision-making in that context. Sharp forecasts are useful to decision-makers only if they are also accurate. Indeed, it is precisely the *lack* of forecast accuracy in current systems that motivates the pooling of hindcasts to explore unobserved parts of the space of drought possibilities in our study. Thus, to address the point, all we can do is reiterate that storylines are not forecasts, and explain that they are instead useful from a disaster risk reduction perspective (rather than e.g. for a cost-benefit optimization exercise as one might undergo based on a probabilistic forecast). The results were presented to the Drought Management Team at Anglian Water (the internal team convened during the 2022 drought), which were able to appropriately consider these results in decision-making. We have moved a sentence from the existing text and added further explanation as follows:

*"It is important to emphasize that the storylines developed here were not meant to be forecasts of winter 2022/23 but instead represented hypotheses of possible river flow and drought responses to explore plausible worst cases. The approach is advantageous from a disaster risk reduction perspective as it increases risk awareness and enables water resources planners to consider a much wider diversity of plausible river flow trajectories. The storylines form one possible source of evidence during drought planning by providing a signal of potentially wet or dry outcomes to plan for." (L387-392 in the revised manuscript).*

The second point raised by the reviewer is whether the storylines are physically realistic since at a long lead time, model solutions may drift from the real world into a physically unrealistic model world. We agree that if the hindcasts contained extreme storylines that were outside those that could be produced in the real world, suboptimal decision-making could result. Thus, establishing the physical plausibility of these storylines is crucial. Yet doing so is inherently challenging since by definition, extreme storylines will lie in the tails of the distributions and will not be well sampled (if at all) in the observations. That is why we conducted model evaluation for our stated purpose, for precipitation (Figures 1 and 2), the circulation drivers (Figure 3), and the atmospheric patterns themselves (Figure 5). Our approach was also compared with the existing ESP method (Figure 7). Further, we have made the following addition to

the text in the methods section to address the reviewer's concern about the potential of exaggerating the likelihood of unrealistically low rainfall:

*"The SEAS5 hindcasts also do not seem to be exaggerating the likelihood of low rainfall as 4 out of the 38 observed winters fall within the lowest 10% of the standardised modelled rainfall distribution. There is a 75% chance of this occurring according to the binomial probability formula." (L105-108 in the revised manuscript)*

We have also added the following text in the discussion:

*"In such an approach it is crucial to establish that the storylines are physically plausible, otherwise decision-making could be suboptimal. Yet doing so is inherently challenging since by definition, extreme storylines will lie in the tails of the distributions and will not be well sampled (if at all) in the observations. That is why we conducted model evaluation for our stated purpose, for precipitation (Figures 1 and 2), the circulation drivers (Figure 3), and the atmospheric patterns themselves (Figure 5). The reliability of the storyline approach ultimately rests on the physical plausibility of the storylines that are produced. However, this is the case for any exploration of extreme outcomes, and is not particular to the storyline approach." (L401-406 in the revised manuscript)*

One section that I did still have a minor objection to is the following (L345-354):

"Although this study did not consider the likelihood of a particular storyline for winter 2022/23, further subsets to the hindcast winters can be made to provide weights for particular storylines that are considered more likely than others over time (e.g. based on prevailing atmospheric circulation patterns). Given the large sample size of the hindcast winters, future work could also condition storylines based on their preconditions. For example, for the 2022 drought, storylines can be created by selecting only winters in the hindcasts with a wetter than average preceding November (as was observed in November 2022). This approach also takes advantage of forecasts of winter circulation characteristics (or weather regimes) which may be more reliable than forecasts of winter precipitation; these circulation forecasts can help inform plausible weightings assigned to particular storylines (Richardson et al. 2020). When employed during an ongoing event, this approach may also shed light on the conditions required for drought termination, for example by calculating drought termination metrics in Parry et al. (2016) for each storyline."

I realise this section is speculating on future improvements, but to me it too strongly retains the sense that the methods presented here are useful in forecasting. I suggest removing this section: as already noted, in my view the methods presented here are not useful for forecasting, and I think the method is unlikely to be useful in prediction, even with the suggestions of future work. This is because the method relies upon ensembles from seasonal climate prediction models, and these models are already more sophisticated alternatives to the inevitable prediction selection etc. required to condition climatological distributions on initial conditions.

**Response:** Although we do see potential in hybrid storyline/forecasting methods, in order to avoid possible confusion we have removed all references to forecasts in this

text and made clear that each of the possible extensions we suggest can be considered as a hypothetical counterfactual, rather than a forecast. The text (now separated off into a separate paragraph) now reads:

*"Although this study did not consider the likelihood of a particular storyline for winter 2022/23, further subsets to the storylines can be made over time or retrospectively. For example, storylines of winter 2022/23 could have been created by selecting only winters in the hindcasts with a wetter than average preceding November (as was observed in November 2022). When employed during an ongoing event, this approach may also shed light on the conditions required for drought termination, for example by calculating drought termination metrics in Parry et al. (2016) for each storyline. The same approach can be used after an event to explore downward counterfactuals and the hydrological impacts should the event have turned out worse. For example, subsets of the storylines can be made to explore consequences should winter 2022/23 have turned out even drier than observed or if the preceding November had been drier than average." (L361-369 in revised manuscript)*